# ROYAL SOCIETY
# OPEN SCIENCE

ecology/environmental science

ecosystem management, population dynamics, wildlife management, seasonal habitat preference, *Sus scrofa*

**Author for correspondence:**
Tadashi Miyashita
e-mail: tmiya@es.a.u-tokyo.ac.jp

# Simultaneous estimation of seasonal population density, habitat preference and catchability of wild boars based on camera data and harvest records

Yuichi Yokoyama[1], Yoshihiro Nakashima[2], Gota Yajima[2] and Tadashi Miyashita[1]

[1]Graduate School of Agriculture and Life Sciences, University of Tokyo, Bunkyo Ward, Tokyo 113-8657, Japan
[2]College of Bioresource Science, Nihon University, Fujisawa, Kanagawa 252-0880, Japan

(iD) TM, 0000-0003-0091-3224

Analyses of life history and population dynamics are essential for effective population control of wild mammals. We developed a model for the simultaneous estimation of seasonal changes in three parameters—population density, habitat preference and trap catchability of target animals—based on camera-trapping data and harvest records. The random encounter and staying time model, with no need for individual recognition, is the core component of the model—by combining this model with the catch-effort model, we estimated density at broad spatial scales and catchability by traps. Here, the wild boar population in central Japan was evaluated as a target population. We found that the estimated population density increased after the birth period and then decreased until the next birth period, mainly due to harvesting. Habitat preference changed seasonally, but forests having abandoned fields nearby were generally preferred throughout the season. These patterns can be explained by patterns of food availability and resting or nesting sites. Catchability by traps also changed seasonally, with relatively high values in the winter, which probably reflected changes in the attractiveness of the trap bait due to activity changes in response to food scarcity. Based on these results, we proposed an effective trapping strategy for wild boars, and discussed the applicability of our model to more general conservation and management issues.

# 1. Introduction

Excessive increases in some mammal populations (e.g. deer, wild boar) have a wide range of negative consequences, such as degradation of ecosystems [1], agricultural damage [2,3] and disease transmission [4]. Implementation of effective management strategies for wildlife populations requires detailed knowledge on the life history and population dynamics of the target species [5–8]. However, mammals in the wild are often cryptic with rare sightings; even their field signs such as footprints and faeces are often difficult to find. Thus, developing a methodological framework for estimating population density of cryptic mammals in a range of spatial scales is necessary.

Camera traps represent a potential alternative for the reliable estimation of the absolute density of cryptic animals. In particular, recently developed approaches to estimate animal density without individual recognition might be applicable. Rowcliffe *et al.* [9] presented the random encounter model (REM) based on ideal gas models [10]. Nakashima *et al.* [11] improved the feasibility of REM by developing the random encounter and staying time (REST) model, in which all required parameters such as population density are estimated exclusively by camera-trap data. More recently, Nakashima *et al.* [12] have incorporated habitat covariates into the REST model, allowing for the likelihood-based estimation of the relationship between habitats and animal density. Since earlier methods assumed simple situations with no spatial structures of densities, this approach yields reliable density estimates at the landscape scale, accounting for spatial and temporal heterogeneity in animal detectability.

The wild boar (*Sus scrofa*) is a cryptic animal species [13,14], which has a broad geographical range with various types of land cover [15], including peri-urban areas [16]. It causes crop damage [17–19] and spreads disease to livestock [20,21]. Although population density and seasonal dynamics are key factors for the effective management of wild boar [22], conventional methods for estimating their numbers are not sufficiently precise or accurate and therefore a standard methodological framework is lacking [23]. Estimating wild boar population sizes by direct observation is difficult due to the preference for dense understorey shrubs and nocturnal and cautious behaviours [24,25]. Indirect indices, such as harvest records (shooting and trapping data) or pellet counts, have been used to estimate population density (e.g. [26–28]), but these indices can be affected by several factors, including season, weather conditions and visibility in habitats [23]. They are thus context-dependent and not reliable due to seasonal and spatial variation.

The spread of wild boar populations has caused serious crop damage all over the world [17,29]. In Japan, wild boars accounted for about 30% of the total crop damage caused by wildlife in 2018 [30]. With the decline in agriculture due to human population ageing in rural areas, abandoned fields and unmanaged bamboo forests preferred by wild boars have been increasing. This has led to crop damage near residential areas [31]. Moreover, the hunter population is ageing and decreasing [32,33], emphasizing the need for higher trap catchability of wild boars with less effort. As noted earlier, population size and habitat use are likely to change spatially and seasonally within a given year, due to the high fertility and omnivorous diet of wild boars [34]. Catchability by traps is also affected by both surrounding land-use and season [35,36], but trapping is based on empirical or anecdotal knowledge, as opposed to an informed and systematically designed approach. Thus, to establish efficient management strategies for wild boar populations, spatial and temporal patterns of population densities and trap catchability should be clarified at the landscape level.

We developed a model for the simultaneous estimation of seasonal changes in three parameters—population density, habitat preference and trap catchability of wild boars—by integrating the REST model and harvest records (figure 1; see Material and methods for details). Simultaneous estimation in a Bayesian framework results in less error than that associated with independent estimates [37], because independent estimates can amplify errors by themselves. Here, the case study was conducted in the southern part of Chiba prefecture, central Japan, where an isolated wild boar population has been increasing since the 1980s, causing severe crop damage [38]. First, to estimate local density and seasonal habitat use, landscape variables were incorporated into the REST model. Second, using estimated landscape parameters and local density, we estimated population density at a large spatial scale, at which harvest records collected by municipal governments are available (mainly the number of trapped individuals and trapping effort). This allowed us to estimate catchability by box and snare traps in each month. Based on the seasonal dynamics of the three parameters, we proposed an effective trapping strategy for sites and timing of trap placement at a landscape level. We also discussed the applicability of our model to more general conservation and management issues far beyond the case study of wild boars.

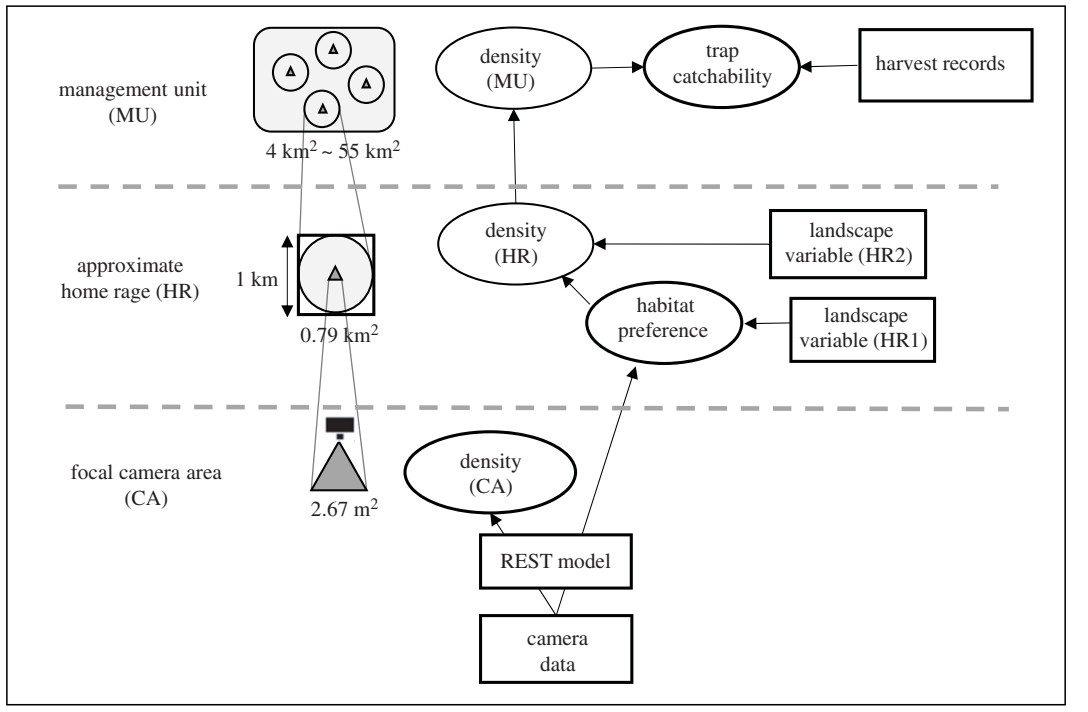

**Figure 1.** General framework of our modelling procedure, which includes estimated parameters (ovals) and data (rectangles) at three different spatial scales (camera site, approximate home range and management unit). For landscape variables at the intermediate scale, two slightly different ones were obtained from land-use data, as habitat preference was estimated in a circle buffer with radius 500 m (HR1), while density was estimated in a 1 km grid scale (HR2). Note that this framework was applied to each month throughout the year.

# 2. Material and methods

## 2.1. Study area and species

The study area included four municipalities in Chiba prefecture, in eastern Japan (35°N, 140°E). The total size of the study area was 799.6 km$^2$, occupying about 15.5% of the prefecture. The climate was classified as warm temperate, with a mean monthly temperature of 4–25°C and annual precipitation of 2000–2400 mm. Forests, including broad-leaf forests (dominated by *Castanopsis sieboldii* and *Quercus* spp. with high acorn production) and conifer plantations (*Cryptomeria japonica* and *Chamaecyparis obtusa*), accounted for about 60% of the area, and farmlands composed mainly of paddy fields accounted for 20% of the study area [39]. Owing to the high browsing pressure by sika deer, the forest understorey vegetation is generally sparse [40], except for abandoned fields where dense shrubs dominate. Bamboo forests, which provide food for wild boars, were also common (3% of the study area). The landscape has a mosaic structure composed of different types of forests, farmlands and residential areas, with severe crop damage by wild boars from adjacent landscapes.

Crop damage caused by wild boars in the study area was about 730 000 USD in 2018 [41]. About 11 300 wild boar individuals were removed by permitted trapping and shooting in 2018 in the study area, with a peak in late autumn to winter (electronic supplementary material, figure S1). The hunting season lasts from 15 November to 15 February, and the number of shot individuals accounted for about 3% of all individuals captured, indicating that trapping accounts for the majority of captures. We, therefore, used only trapping data for the subsequent analyses, which are hereafter referred to as harvest record.

Most female wild boars in Japan give birth to a litter of about four piglets on average during May and June [42]; accordingly, the population size of wild boars can fluctuate considerably even within a year. Due to their omnivorous diet, wild boars are known to change their habitat use seasonally, depending on food availability [17,43]. Especially in Japan, wild boars eat various kinds of foods, such as bamboo shoots, oak mast and grasses, and the food items change according to season [31,44].

In this study, juveniles were defined based on a distinct stripe pattern by camera observation. This stripe pattern is lost at four months after birth, when the body weight exceeds 15 kg [45]. From

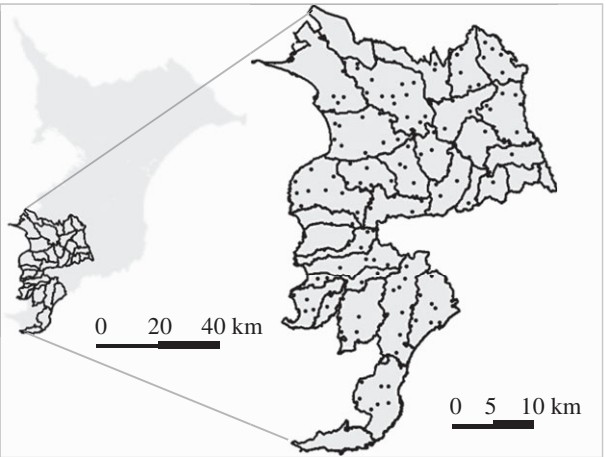

**Figure 2.** The study area consisting of 33 management units of wild boars (areas enclosed with lines) and the locations where camera traps were placed (dots).

camera observation, yearlings without stripe patterns are indistinguishable from adults, so only the two age groups (juvenile and adult) were used in this study.

## 2.2. Data collection

In total, 180 motion-activated camera traps (Strike Force HD Pro; Browning Trail Cameras, Morgan, UT) were randomly placed in forests (broad-leaved, conifer plantation or bamboo) in the study area. Although the wild boar also uses abandoned fields [31], cameras were not placed in these areas to avoid the infringement of privacy rights (i.e. to avoid obtaining images of people in the areas). Camera density was about one camera/4.4 km$^2$ (figure 2). Camera records were obtained from June 2018 to May 2019. Each camera was set on a tree trunk about 1 m above the ground. An infrared flash enabled the camera to capture images of animals passing in front of it, even at night. All cameras captured 20 s videos with a 1 s delay between consecutive images. About every three months, the SD card and battery were replaced in all cameras.

The landscape variable was based on land-use data within a 500 m fixed-radius buffer around each camera, which was used to estimate habitat preference of wild boars. This buffer size was determined by referring to the average home range size of wild boars reported in Japan [46]. Thus, the spatial scale of habitat preference estimated here represents approximate the home range size (figure 1). Six types of landscape elements were extracted from a national survey carried out from 2000 to 2008 [39]: broad-leaf forest, conifer plantation forests, farmland, bamboo forest, abandoned area and residential area. Using ArcGIS 10.6 (Esri Inc., New York, NY), the percentage of each landscape element within each buffer was calculated. All variables were standardized (mean = 0, s.d. = 1) prior to analysis. To eliminate multicollinearity, six landscape variables were summarized by principal component analysis (PCA). Principal component axes 1–4 were used as landscape variables that may influence wild boar ecology (see Results for PCA axes).

Harvest records were provided by the governments of Chiba Prefecture, Futtsu City and Minamiboso City. These data included numbers of active traps and the details of individual animals trapped (e.g. date, sex, body weight, type of trap). Numbers of traps were recorded at the level of 'management unit' defined by the municipal government (figure 2), with management units ranging from 4 to 55 km$^2$. Numbers of active traps during a month were used as trapping effort. Individuals weighing less than 15 kg with juvenile-specific stripe patterns were defined as juveniles; otherwise, individuals were classified as adults [44]. Yearlings were not distinguished to match the classification of camera data.

Two types of traps were used in the study area: box traps (electronic supplementary material, figure S2: left side) and snare traps (electronic supplementary material, figure S2: right side). Box traps contained bait such as rice bran to attract wild boars. Snare traps consisted of a wire and spring and were buried underground. When an individual stepped on a snare trap, the wire entangled a leg. In the study area, about 72% of all captured wild boars were trapped by box traps, 25% were trapped by snare traps, and the remaining percentage was gun hunted.

## 2.3. Analysis

Seasonal changes in population density, habitat use and trap catchability were estimated by a series of equations described later using a Bayesian framework (figure 1).

### 2.3.1. Local population density and habitat use

Using the REST model [11], wild boar density was estimated from camera-trap data. The REST model is an extension of the REM [9]. The REM assumes that the detection rate is a function of animal density, animal movement speed and the detection area of a camera; movement speed needs to be estimated from an alternative data source, such as radio telemetry. However, the REST model uses staying time in a predetermined area within view of a camera (hereafter, staying time), instead of movement speed. Staying time is inversely proportional to movement speed and is easily measured from camera recordings. To fix the camera detection area, an equilateral triangle with a side length of 1.9 m within the view of a camera in the field was set (electronic supplementary material, figure S3). We assumed that all animals passing through the focal area could be detected, like Nakashima *et al.* [11]. Passages by wild boars were counted, and the staying time at each passage was measured following the protocol described by Nakashima *et al.* [11].

To estimate wild boar density by the REST model, the following data were extracted from camera records in each month: (i) number of passages through the focal area, and (ii) staying time within the focal area. Density was estimated by the following equation:

$$[\text{density}]_{i,m} = \frac{[\text{numbers of passage through focal area}]_{i,m}}{[\text{area of focal area}]} \times \frac{[\text{staying time estimates}]_m}{[\text{active time}]_m},$$ (2.1)

where $i$ indicates the camera ID ($i = 1, 2, \ldots ,180$) and $m$ indicates the month ($m = 1, 2, \ldots ,12$). Because wild boar activity is assumed to change seasonally, the staying time within the focal area was measured for each month. About 50 videos were selected from several sites each month, and staying time was measured using a stopwatch in a laboratory. To estimate the average staying time for each month, staying time data were fit to four distributions (exponential, gamma, lognormal and Weibull distributions), and the best-fit model was determined by the Watanabe–Akaike information criterion (WAIC) [47]. We also tested whether the incorporation of random effects at camera station improve the model predictability based on the WAIC value. As a result, the lognormal distribution without the random effects fitted the data the best (see electronic supplementary material, table S1). The staying time of juveniles and adults was estimated separately (see electronic supplementary material, figure S4). The density of juveniles was estimated only from June to October in 2018 and in May in 2019 because the counts were too small to estimate staying time in other months. The 'active time' in equation (2.1) was calculated from the total recording period multiplied by the 'daily activity proportion of time'. The daily activity proportion of time was estimated as the daily change in the number of times wild boars were recorded by the camera [48], or the proportion of active time in a 24 h period. The REST model assumes that all individuals are active at the peak of the activity rhythm [11]. Because the daily activity proportion may differ among months and ages, separate estimates were obtained for juveniles and adults each month (see electronic supplementary material, figure S5). The negative binomial distribution (i.e. the Poisson distribution with random effects following a gamma distribution) was fitted to the number of passages through the focal area, as in Nakashima *et al.* [11]. The details of the REST model and analysis are described in Nakashima *et al.* [11].

Seasonal habitat preference was assessed by incorporating landscape variables (summarized as PCA axes) as covariates into the REST model, as expressed in equation (2.2). The coefficient $\alpha$ associated with landscape variables was used as the index of habitat preference in the home range, as $\alpha$ indicates fine-scale spatio-temporal variation in wild boar densities. The following equation was used to assess habitat use:

$$[\text{density}]_{i,m} = \exp\left(\alpha_{0,m} + \sum_{e=1}^{4} \alpha_{e,m} \times [\text{PC}]_{e,i}\right),$$ (2.2)

where $i$ indicates the camera ID ($s = 1, 2, \ldots ,180$), $m$ indicates the month ($m = 1, 2, \ldots ,12$) and $e$ indicates the type of landscape variable. PC indicates the principal component scores summarizing landscape variables around each camera, $\alpha_0$ represents for the intercept (average density across environments) and $\alpha_1$ to $\alpha_4$ represent how each landscape composition affects the local wild boar density (i.e. the habitat preference in each month).

From June 2018 to May 2019, we observed 2191 passages of adults and 1306 passages of juveniles within the focal area, and these data were used for the REST model.

### 2.3.2. Population density at the management-unit level

To estimate the wild boar density at the management-unit level, densities in 1 km$^2$ grid cells over the whole study area were estimated by equation (2.3), assuming that landscape variables determine density,

$$[\text{density}]_{c,m} = \exp\left(\alpha_{0,m} + \sum_{e=1}^{4} \alpha_{e,m} \times [\text{PC}]_{e,c}\right), \tag{2.3}$$

where $c$ indicates a 1 km$^2$ cell ($c = 1, 2, \ldots, 897$) and $m$ indicates the month. Principal component scores for the 1 km$^2$ cells were calculated by the percentages of landscape elements at each cell, combined with PCA axes in equation (2.2). Second, the density at the management-unit level was estimated by calculating the weighted average of cell densities, with the weights based on the forested area in each cell.

### 2.3.3. Catchability

Catchability was estimated by harvest records of wild boars and the estimated parameters in equations (2.1) and (2.2). We assumed that the number of captured individuals was proportional to trap effort and animal density, and the proportionality constant was defined as trap catchability, as in Osada *et al.* [28]. The following equation was used to estimate catchability:

$$[\text{trapped number}]_{u,m,t} | [\text{average of trapped number}]_{u,m,t} \sim \text{poisson}([\text{average of trapped number}]_{u,m,t})$$

$$[\text{catchability}]_{u,m,t} = \frac{[\text{average of trapped number}]_{u,m,t}}{[\text{the number of traps}]_{u,m} \times [\text{density}]_{u,m}}, \tag{2.4}$$

where $u$ indicates the management unit defined by the municipal government ($u = 1, 2, \ldots, 33$), $m$ indicates the month ($m = 1, 2, \ldots, 12$) and $t$ indicates the type of trap ($t = 1, 2$; 1 indicates box trap and 2 indicates snare trap). The number of trapped individuals and the number of traps were obtained from harvest records provided by municipal governments.

We constructed a hierarchical Bayesian model including equations (2.1–2.4) to estimate parameters simultaneously. Prior distributions for $\alpha$ were uniform (−3, 3), and those for catchability parameters were uniform (0, 1). Posterior samples and distributions were obtained by MCMC sampling (see electronic supplementary material, figure S6). The number of MC iterations was 50 000, the burn-in was 10 000 iterations, and the thinning interval was 20. The number of chains was 3. The convergence of the MCMC samples for all estimates was checked using Rhat (less than 1.1; [49]). This analysis was performed using JAGS [50] with R (v. 3.5.0, [51]). The JAGS code and settings are provided in 'R code' in the electronic supplementary material.

## 3. Results

The PCA axes summarizing landscape variables surrounding camera sites set in forests are shown in figure 3. PC1 was positively associated with forests and negatively associated with the other landscape elements. PC2 represented a gradient of broad-leaved forests versus conifer plantation forests, with positive scores for plantations and negative scores for broad-leaf forests. PC3 was positively associated with abandoned fields. PC4 represented mainly bamboo forests. The proportions of landscape variation explained by the four axes were 36%, 23%, 17% and 13%, respectively, for axes 1–4.

### 3.1. Population density

Seasonal patterns in population densities differed between adult and juvenile wild boars (figure 4). The density of adults increased gradually from July to November and decreased thereafter. The density of juveniles increased in May and gradually decreased until October. This contrasting pattern may reflect the birth season in May in Japan [42] and the subsequent high mortality of juveniles and recruitment of juveniles into the adult stage, which may increase the adult population.

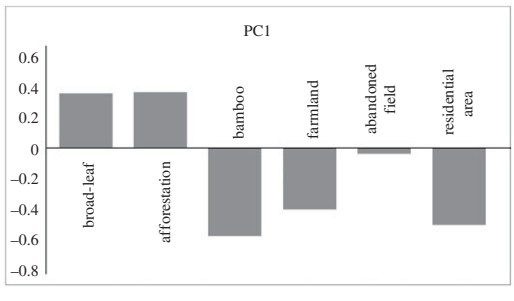
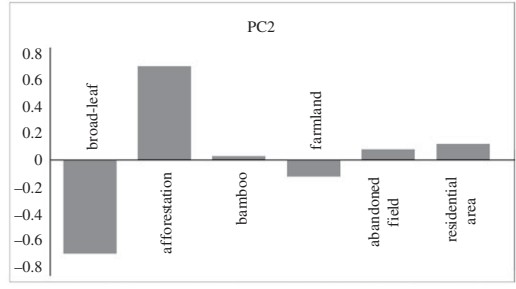
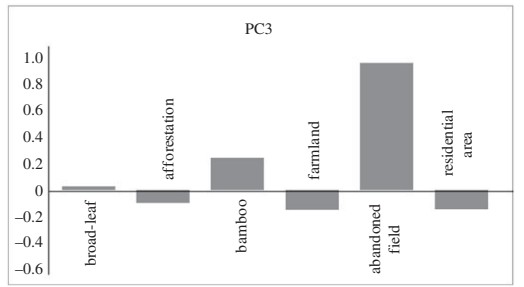
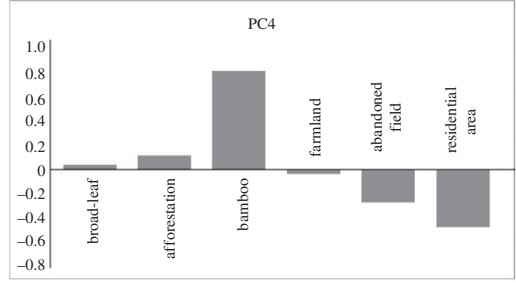

**Figure 3.** Factor loadings of each landscape element that constitutes principal component axes.

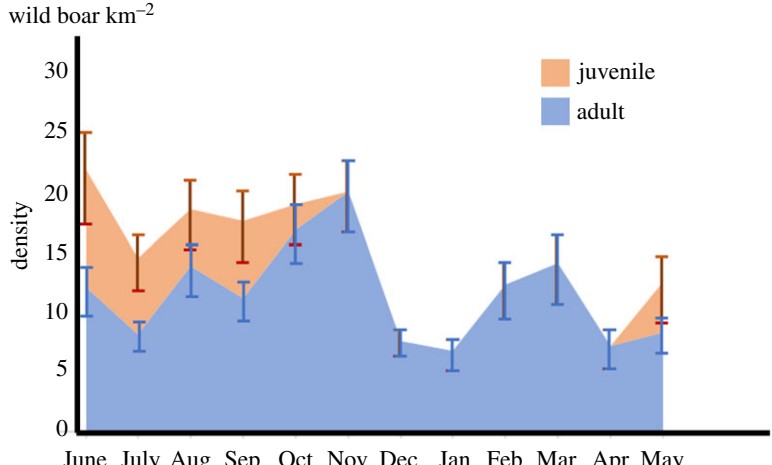

**Figure 4.** Seasonal changes in the estimated densities of juveniles (orange area) and adults (blue area). Whiskers represent the 50% credible interval. Note that this is a stacked area graph.

## 3.2. Habitat preference

The habitat preference of wild boars was assessed from the coefficients associated with landscape variables around camera traps (figure 5). The coefficient score for PC1 was mostly negative throughout the year, indicating a preference for landscapes with farmlands and bamboo forests. The coefficient for PC2 was also negative for most of the year, with strongly negative values from October to November and from April to May, indicating a preference for broad-leaved forests over conifer plantation forests, particularly in the periods from October to November and from April to May. The coefficient score for PC3 was generally positive throughout the year, with relatively large values in June and from December to February, indicating a particularly strong preference for abandoned fields during these periods. The coefficient for PC4, representing mainly bamboo forests, was positive throughout the year, but the effect was weaker than those of other PC axes.

## 3.3. Catchability

Catchability was estimated from population density and harvest records at each management unit. Catchability by box traps was about 1.7 times higher and more variable than that by snare traps (figure 6).

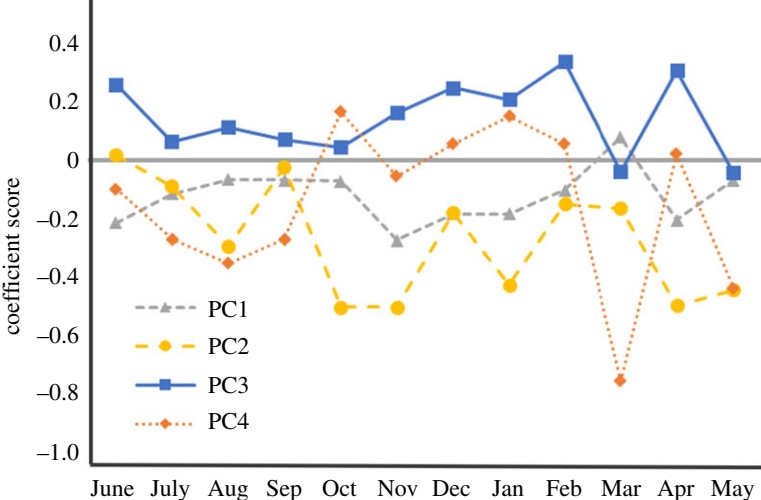

**Figure 5.** Estimated values of coefficients associated with the environmental variables. Filled symbols indicate coefficients whose 95% credible intervals do not overlap 0.

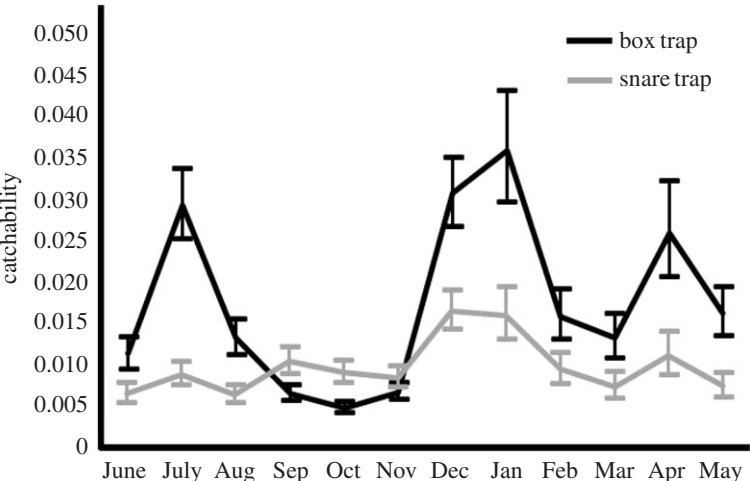

**Figure 6.** Seasonal change in estimated catchability of two types of traps (box and snare traps) in each month. Whiskers represent the 50% credible interval.

The catchability by both types of traps showed clear seasonal patterns. A common feature is the increase in catchability from December to January (figure 6). From August to November, the trends differed between box and snare traps. The catchability of box traps remained relatively low in this season, while that by snare traps increased over time.

## 4. Discussion

Using a Bayesian framework, we achieved the simultaneous estimation of seasonal dynamics of population density, habitat preference and trapping catchability of wild boars. These parameters are useful for wildlife management, including the efficient spatial and temporal allocation of trapping effort, assessment of the effectiveness of population control programmes, and habitat management to reduce crop damage. The simultaneous estimation of these three parameters (i.e. population density, habitat preference and trapping catchability) has a substantial advantage over separate estimates with respect to accuracy and precision. The statistical model that we developed uses two different data types with contrasting spatial resolution; camera records are locally precise, while harvest records provided by municipal governments can cover a broad spatial scale. This improves the reliability of parameter estimates.

Considering seasonal fluctuations, we seemed to have obtained reasonable estimates of population density. The maximum density estimated in our study was in agreement, albeit slightly lower than in a

previous estimate for the same area of 26 km$^{-2}$ after the population increase within a season and before harvesting [28]. Moreover, the difference in seasonal density patterns between adults and juveniles can generally be explained by differences in life history and harvesting patterns in the study area. Females give birth in May and June [42], juveniles grow and lose their stripes in about four months [45] and mortality by harvesting increases in the winter (see electronic supplementary material, figure S1). There are, however, two points that should be addressed. First, the rapid density decrease after November cannot be explained solely by harvest mortality. It can potentially be explained by the shift in habitat preferences from forests to abandoned fields, as demonstrated by the increase in the PC3 coefficients (figure 5). This seems plausible because abandoned fields are known habitats for wild boars in Japan [31]. Because all cameras were set in forests, density estimates in forests may have decreased in this season. Another point to note is that estimated juvenile densities were lower than those of adults, which should be an underestimate. On average, most female wild boars in Japan give birth to a litter of about four piglets on average [42]. With the assumption of an even sex ratio, the density of juveniles should be about twice that of adults just after the breeding season. The underestimation could be explained by the behaviour of juveniles. Soon after birth, juveniles are cared for by females around the resting sites [34], while females spend time on foraging activities because lactation increases energetic needs [52]. Due to frequent resting by juveniles, the assumption of the REST model that all individuals are active at the peak of the activity rhythm could be violated. This might lead to the underestimation of the daily activity and thereby juvenile densities.

The detected habitat preferences also seemed to accurately reflect the feeding and behavioural habits of wild boars. We found that wild boars prefer broad-leaf forests, abandoned fields and bamboo forests to varying degrees. The preference for broad-leaf forests was rather high and became stronger in the autumn and in spring. Conifer plantations (*C. japonica* and *C. obtusa*) are known to be less preferred by wild boars due to the lack of food resources [31,53], while broad-leaf forests are good feeding sites for the species [43,53]. In the study area, broad-leaf forests consisting mainly of Fagaceae trees produce a large number of acorns in the late autumn, which are highly digestible and nutritional for wild boars [54]. Moreover, soil invertebrates such as earthworms become abundant in broad-leaf forests from April to May [55]. Such food phenology may explain the seasonal dynamics observed in this study.

We found the effects of various surrounding landscape elements on local densities in forests. In particular, farmlands, abandoned fields and bamboo forests generally enhanced the local density of wild boars in forests. These landscape elements seemed to provide wild boars with seasonal resources. For instance, we detected a preference for bamboo forests from December to February. In Chiba prefecture, wild boars consume bamboo shoots in the winter [56]. In addition, we detected a strong preference for abandoned fields in June and December–February. Abandoned fields provide feeding and resting sites to wild boars in Japan [31,44], including above-ground plant materials from the spring to summer [57] and roots in the winter [44]. Furthermore, the hunting season from 15 November to 15 February is likely to promote the use of abandoned fields as refuges [58]. Because the understorey vegetation in forests is highly browsed by sika deer in our study area [40], wild boars may use abandoned fields with dense shrubs as resting sites, rather than forests with sparse vegetation.

We also clearly detected seasonal trends in trap catchability, as reported in earlier studies [35]. One prominent feature was an increase in catchability in the winter, which was common to both box and snare traps. This can be attributed to the food shortage in the winter, when there are no crops on farmlands and the nutritional condition of wild boars is poor [57]. A food shortage leads to an extension of the home range to search for foods [59], which may enhance the capture probability. Increased attraction to trap bait due to food shortage may also explain the higher catchability in winter, but this applies only to box traps. Contrary to the common trend in the winter, the catchability trends from August to November were different between box and snare traps. Catchability by box traps in July was rather high and decreased thereafter, which could be due to abundant food resources. In particular, acorns in broad-leaf forests are abundant in the autumn. Because acorn availability is negatively correlated with the consumption of other resources [60], wild boars are less likely to be attracted by baits in this season. Catchability by snare traps showed a different pattern; it increased slightly from June to November. The reason for this trend is not clear, and further work is needed to determine whether it is significant and meaningful.

## 4.1. Management implications

We developed a novel statistical framework using camera trapping and harvest records to uncover seasonal changes in the population density, habitat preference and trapping catchability of wild boars in Japan. Most of the observed changes could be explained by the dynamics of food availability in landscapes, the reproductive

cycle and harvesting by humans. These results can help hunters and local policymakers determine effective seasons and landscape types for trap placement and for gun hunting.

Our results provide a basis for specific recommendations for the management of wild boar populations. Trapping effort should be increased in the winter when catchability is high. In this season, wild boars showed relatively high utilization of abandoned fields; accordingly, setting box traps around abandoned fields, with daily inspection, should be prioritized. In other seasons, trapping locations can be determined based on wild boar habitat selection. For instance, from September to October, the main habitat shifts from abandoned fields to broad-leaf forests, and the opposite directional shift is found from November to December. The seasonal patterns of habitat use and catchability may be regional and hence are not generalizable. However, such information can be obtained by the application of our model in cases where camera-trapping data and harvest records by municipal governments are available.

Our model used both camera and trapping data to estimate density in a range of spatial scales, but the essence of our model is to integrate REST model (using camera data) and catch-effort model. Consequently, it can be readily used for any types of hunting data, including ground and aerial shootings, if hunting efforts are available. Furthermore, our modelling framework can be applied to various situations, including endangered species conservation and invasive alien management. For instance, wildlife populations in tropical forests are often cryptic and hunted by local people for bushmeat, and scientifically based guidelines that enable the balance of their conservation and utilization are urgently required [61,62]. Earlier studies estimated population densities with single density proxy or multiple proxies used separately [61,62]. But by combining hunting records over a broader spatial scale with high-resolution camera data, it is possible to know more unbiased population estimations. Other than species conservation, our models can be used for eradication programmes of invasive aliens, in which catch-effort model and camera-trapping model were separately used in earlier studies [63–65]. Finally, camera-trapping data spanning several years make it possible to estimate demographic parameters (such as survival rates) and long-term population dynamics. These technical advances will make a substantial contribution to the management of wildlife that are difficult to detect by traditional methods.

Ethics. Harvesting records used in this study were provided by local governments in Chiba prefecture, and the harvesting was conducted by the local governments under Wildlife Protection and Hunting Management Law in Japan.

Data accessibility. The datasets supporting this article is available in the electronic supplementary material.

Authors' contributions. Y.Y. carried out data analysis and drafted the manuscript; Y.N. participated in the design of the study, collected field data and helped data analysis; G.Y. collected field data and analysed video recordings; T.M. participated in the design of the study and critically revised the manuscript. All authors gave final approval for publication and agree to be held accountable for the work performed therein.

Competing interests. We have no competing interest.

Funding. This study was financially supported by the Grant-in-Aid for JSPS Fellows (KAKENHI grant no. 17H01916) and the Environmental Technology Development Fund, Ministry for the Environment, Japan (SUISHINHI grant no. 4-1704).

Acknowledgements. We thank the public officers of the municipal governments in our study area. We are also grateful to H. Yokomizo, G. Takimoto and G. Fujita for helpful comments. Data collection was supported by students of Nihon University College of Bioresource Science. We appreciate their huge efforts.

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
