## [Reviewer comments · Royal Society Open Science]

Review History

RSOS-200579.R0 (Original submission)

Review form: Reviewer 1

Is the manuscript scientifically sound in its present form?

Yes

Are the interpretations and conclusions justified by the results?

Yes

Is the language acceptable?

Yes

Do you have any ethical concerns with this paper?

No

Have you any concerns about statistical analyses in this paper?

No

Recommendation?

Accept with minor revision (please list in comments)

Comments to the Author(s)

The authors developed a model for simultaneous estimation of seasonal changes in different parameters for an example wild boar population in Japan. The model is a combination of the random encounter model and the staying time model and includes harvest records and trap cam data to estimate wild boar density on a broad spatial and seasonal scale.

It is an interesting manuscript with a timely topic. The topic is very important from the management point of view, because wild boar can impose heavy impacts on biodiversity and human activities. I think that this manuscript can be a major contribution to the research field. The manuscript is also in general well-structured and clear.

Still, I think that there are some revisions required.

I start with some major concerns:

1. I think that this manuscript is written as a standard research paper right now but has the potential to be written up as a more methodological paper. At the moment the authors mention that they introduce a novel statistical framework, but as reader you do not get the feeling that this is the main focus. I would advise restructuring the manuscript and put the main focus on the method development and using the wild boar population as a case study. Annotations in the attached r-script can help to make it reproducible for researchers dealing with similar topics/challenges. That will also involve that the advantages (“simultaneous estimation”) of this novel statistical framework get more attention. The authors mention it, but in my opinion it misses some depth and details. It seems that the simultaneous estimation is one major advantage, but it gets a bit lost in the biological interpretation of the case study (wild boar population). I think that by restructuring it to a “methodological paper”, the authors can reach a much broader audience of researchers dealing with harvest species in general.
2. This leads me directly to the next feedback point. At the moment the authors show that this “novel statistical framework” can be used for harvest species. But couldn't they expand/generalize it for species that need to be protected? The authors could also reach conservationists by showing how this method could be used for an endangered species for instance. Proper and reliable density estimation are also of huge importance for endangered species.
3. I am a bit concerned to what extent this model can be used by managers/researcher that do not have harvest records/trapping data, but only rely on camera trap data for instance. It seems that in Japan trapping records are important measure to control the population of wild boar, however, in Europe, as I know, hunting plays most important role and trapping is not allowed to control the population. That makes it difficult to translate this approach to a European situation. I would advise that the authors elaborate this issue (challenges and limitations for generalization of the statistical framework) in a separate section of the manuscript. In addition, adding information about a more general use will make it much more interesting to a broader public. I also miss this aspect in the last section of the manuscript about management implications. Can this model be used if no trapping data are available? How would I need to deal with it?
4. One confusing thing for the reader is that the authors switch between the terms “harvest record” and “trapping records”. They need to be consistent with it to not confuse the reader. In addition, it would be good to explain “harvest record” already shortly in the introduction, since it also appears in figure 1 (in the introduction). To me the difference between hunting bag and harvest record was not clear in the beginning. Defining harvest record will help the reader a lot.

5. I am a bit concerned about the fact that the authors did not include more age classes. They used only juveniles (<4 months) and adults (>4 months). As mentioned in my specific comments below, I would add yearlings because we cannot be certain that adults and yearlings do not differ in their habitat use, activity pattern and catchability.

6. I would also recommend to the authors to make clearer in the result section that the analysis were conducted on different spatial scales. At the moment it seems that some interpretations from the results are not correct, however, after reading carefully the methods again it becomes clear. As an example. In the summary, the authors mention the habitat preference of forests adjacent to abandoned fields. From the result section, this conclusion does not become really clear, still I am certain that the analysis were done on this.

7. A bit a minor remark, but since it is a general remark and not mentioned in specific minor comments: In general I would rather refer to "landscape variables" than "environmental variables", since environmental can be much broader and are less specific. Landscape gives the reader a more concrete idea of what is meant.

You find my specific (minor) comments below. I hope they be useful

Minor comments:

Page 2:

L40: instead of ";" – sign, I would advise a point as end of the sentence.

L44: evaluated "as an example population". I would add this to make it clearer for the reader.

L45: mainly due to harvesting

L45-L46: I do not see where this conclusion/result come from in your result section. In your results, you do not say that forests adjacent to abandoned fields are preferred and I don't think that you analysed this. If you did so, please make it clear to the reader.

L47: so throughout the season? I would add this in the sentence to make it clearer to the reader.

L50: for me it sounds that attractiveness of trap bait and activity changes in response to food scarcity is similar or at least closely related. I would rather write: ..., which probably reflected changes in the attractiveness of the trap bait due to activity changes in response to food scarcity.

Page 3:

L3: I advise putting keywords in alphabetical order.

L9: I would recommend mentioning some example mammal populations in brackets.

L16: I think I understand what you mean by field signs but maybe short elaboration is better.

Give an example for instance. I am not sure if field sign is the most appropriate word here.

L27: what is the difference between hunting bag and harvest record? I would have expected that it is the same. Thus, I would advise explaining the definition of harvest record already in an early stage since it is also used in the title.

L41: can you elaborate a bit more which parameters for instance are estimated by the camera trap data so that it is better to understand for a reader without the background knowledge of the REST model?

L51: You can even put it in a broader context. Not only in Japan but worldwide wild boar are causing damage. You could start with the fact that worldwide wild boar are causing damage with some countries as example and then narrow down to also in Japan they are causing damage and continue your story. That makes it maybe even more interesting for a broader public.

L52: "As abandonment of agricultural fields increases due to human population aging, sites preferred by wild boar, including abandoned fields and bamboo forests, have been increasing, thereby leading to crop damage near residential areas. This sentence is not really clear to me. It is a long sentence and a bit confusing. I would recommend rephrasing it.

Page 4:

L2: delete "the" in front of "environmental conditions".

L2-L9: that paragraph is a bit confusing to me. I think the main points you want to make is that catchability, population size as well as habitat use change spatially and seasonally. Because you

mention the reason for two of the factors (habitat use and population size) in between of the enumeration of the three parameters the readers gets a bit confused. I would advise rephrasing this paragraph.

L17: Why does simultaneous estimation in a Bayesian framework results in less error? I would recommend giving some additional explanation for this. Does not need to be long, but one addition sentence may help to get more clarity here.

L20: You mention here trapping records. I guess that this is the same as harvest records? But that is a guess and not really clear. I would advise using consistent words for it. And also some additional information about it. I was wondering for instance how sure you are about the fact that trapping by different municipalities and during different seasons was conducted with a constant effort? If that is an assumption you have to be quite certain about it, because I think that this can have huge impact on the results if it is not conducted with constant effort. Otherwise it might be really similar to hunting bag? I see now that you mention the trapping effort in the methods. That's good, but I would say, make sure that the reader knows here already that you take different trapping effort into account. Simply one short additional sentence should be enough.

L24-L29: I would rephrase this last part of the introduction. At the moment it feels a bit lost from the earlier paragraph. I would recommend something like: "Based on seasonal dynamics of the three parameters, we proposed an effective trapping strategy for sites and timing of trap placement on a landscape level for an example study area in the southern part of Chiba prefecture, central Japan." By simply switching first and last part of your paragraph it better links to the earlier paragraph.

Page 5:

L1: There seems to be a clear distinction between hunting and trapping. I think that this needs to be clearer for the reader at an early stage. I start now understanding that there is a difference, but it can be much easier for the reader if it is explained clearly at the beginning of the manuscript. I think that it is not so common to have trapping (as you define it) next to hunting methods for wild boar control. At least not in Europe. Thus, I guess that European readers may be a bit confused here.

L6-L11: I think that you can better combine those two sentences. Now it seems a bit that you rephrase the same sentence again.

L14: If I understand correctly you distinguished between juveniles (0-4 months) and adults (>4 months). Wouldn't you expect differences in catchability and habitat use between piglets (<13 months), sub-adults/yearlings (13-24 months old individuals) and adults (>24 months old individuals) as well? I understand if the distinction of other age classes only on camera traps is not possible but I think that it is important to mention this possible limitation of the study here.

L21: where the camera traps randomly distributed throughout the area? I guess so because you used the Rowcliffe method, but I don't see that you mention it in this paragraph.

L35: "The environmental variable was based on land use data..." □ add "was based" to your sentence.

L37: add "average" home range? That is probably more appropriate.

L45: I think that it is clearer if you write "... six landscape elements..." instead of "six environmental elements".

L60: see my comments on the age classes before. I think some additional explanation is necessary why you did not distinguish between yearlings and adults as well. Especially when you have trapping data more than only the two age classes you distinguish may be available.

Page 6:

L9: replace "rest were" by "remaining percentage was".

L29: what is the distribution of the staying time? is there a lot of variation? I would like to see a histogram of the staying time variable in the supplementary material.

Page 8:

L17: Put posterior distributions in the supplemental material please.

L28: rather than “environmental variables”, I would name it “landscape variables”. That is more precise and clearer to the reader.

L39-41: I would put those two lines in the methods instead of in the results.

Page 9:

L27: “simultaneous estimation”. Is that one of the major advantages of this technique/tool that you used here? If so, I would make that clear to the reader. At the moment it pops up in the title and here in the first sentence of discussion, but to me it is not really clear if that is an important point or not. Needs some clarification here. Ok, I see now in the next sentences of the discussion that you mention the advantages. I would already see those points in the introduction (when elaborating the problem).

L29: delete “quite” in sentence “quite useful”. It is useful!

L40: I miss a bit more in-depth detail about how exactly this methods shows advantages. Why and how are parameter estimates more reliable?

L45: refer to the reports. How do we know what is the “correct” density? How do we know if those estimates are more reliable than others, which are also estimated? That is not explained enough here and should be more convincing.

L60: can you add an overview of number of cameras in each landscape element in the study area (in supplemental material) so that you can underline your word “mainly” by a number/percentage. That makes your sentence more convincing.

Page 10:

L4: ON average. Not “in average”.

Page 11:

L20: Can you also mention management implications for countries that do not use trapping as control measure for the population?

L39-L46: As mentioned in my general comments, you can put much more focus on this last paragraph in your introduction and make it much more interesting for a broader public.

Page 12:

L10: reference nr 17 populationabundance should be two separate words.

Page 15:

Figure 1: What are the boxes and circles indicating? Mention it in the figure description. I anyway need some addition explanation for figure 1. When you only read the introduction where the first time is referred to figure 1 and by having only those information from introduction and figure description it is difficult to understand the figure. For instance, what is meant with density/mesh? I would advise either give additional explanation in introduction and/or figure description or make an easier graphical representation for the introduction.

Page 16:

L 50: Figure 4: is juvenile density never below adult density or not visible in month December-April for the juveniles because it lies below the density of the adults? If the latter is the case I would advise to use lines instead of filling the surface. Wouldn't be nicer to add also the bars in the two different colors (light and dark grey) for the two age classes? What is the reason to use the 50% credible interval and not a 95% confidence interval or a standard error?

Page 17:

L20: put the x-axis with the month at the bottom (at y=-1 for instance) so that it is easier to read and not all the months names are below the lines and points. This makes the graph a bit confusing.

L54: “throughout the year” instead of “through a year”.

Supporting information:

L14: typing error: “monitoring”.

L20: Fig. S4: b) y axis is entitled with number of videos with two most left graphs, but has frequency on y-axis for the two most right and two central graphs. I think that you should remove the "frequency" on the y-axis, as it is also removed in (a). I don't understand in the figure description what you mean with y-values of circular points in the figures. I don't see any circular points.

L26: In reference Rowcliffe et al, (2014) is missing.

Review form: Reviewer 2

Is the manuscript scientifically sound in its present form?

Yes

Are the interpretations and conclusions justified by the results?

Yes

Is the language acceptable?

Yes

Do you have any ethical concerns with this paper?

No

Have you any concerns about statistical analyses in this paper?

Yes

Recommendation?

Major revision is needed (please make suggestions in comments)

Comments to the Author(s)

Dear authors,

I have found your work timely and interesting. My main concern is the potential pseudoreplication in your experimental design. I think that the incorporation of the camera id as a random term (e.g., intercept), will improve your Bayesian modelling preventing from getting spurious effects (see, Paul-Christian Bürkner. 2020. Package 'brms' 2.12.0 version). Other minor comments have been included in the pdf (Appendix A).

I hope that my comments will be useful for your investigation

Yours sincerely

Decision letter (RSOS-200579.R0)

Dear Dr Miyashita,

The editors assigned to your paper ("Simultaneous estimation of seasonal population density, habitat preference, and catchability of wild boars based on camera data and harvest records") have now received comments from reviewers. We would like you to revise your paper in

accordance with the referee and Associate Editor suggestions which can be found below (not including confidential reports to the Editor). Please note this decision does not guarantee eventual acceptance.

Please submit a copy of your revised paper before 19-Jun-2020. Please note that the revision deadline will expire at 00.00am on this date. If we do not hear from you within this time then it will be assumed that the paper has been withdrawn. In exceptional circumstances, extensions may be possible if agreed with the Editorial Office in advance. We do not allow multiple rounds of revision so we urge you to make every effort to fully address all of the comments at this stage. If deemed necessary by the Editors, your manuscript will be sent back to one or more of the original reviewers for assessment. If the original reviewers are not available, we may invite new reviewers.

- Data accessibility

<http://datadryad.org/submit?journalID=RSOS&manu=RSOS-200579>

- Competing interests

- Authors' contributions

All submissions, other than those with a single author, must include an Authors' Contributions section which individually lists the specific contribution of each author. The list of Authors

should meet all of the following criteria; 1) substantial contributions to conception and design, or acquisition of data, or analysis and interpretation of data; 2) drafting the article or revising it critically for important intellectual content; and 3) final approval of the version to be published.

- Acknowledgements

- Funding statement

Associate Editor's comments:

Two reviewers have provided feedback on your work. As there are a number of queries that should be addressed in your revision, please ensure you fully respond to these in your revised manuscript and accompanying response to reviewers document.

Reviewers' Comments to Author:

Reviewer: 1

Comments to the Author(s)

The authors developed a model for simultaneous estimation of seasonal changes in different parameters for an example wild boar population in Japan. The model is a combination of the random encounter model and the staying time model and includes harvest records and trap cam data to estimate wild boar density on a broad spatial and seasonal scale.

It is an interesting manuscript with a timely topic. The topic is very important from the management point of view, because wild boar can impose heavy impacts on biodiversity and human activities. I think that this manuscript can be a major contribution to the research field. The manuscript is also in general well-structured and clear.

Still, I think that there are some revisions required.

I start with some major concerns:

1. I think that this manuscript is written as a standard research paper right now but has the potential to be written up as a more methodological paper. At the moment the authors mention that they introduce a novel statistical framework, but as reader you do not get the feeling that this is the main focus. I would advise restructuring the manuscript and put the main focus on the method development and using the wild boar population as a case study. Annotations in the attached r-script can help to make it reproducible for researchers dealing with similar topics/challenges. That will also involve that the advantages (“simultaneous estimation”) of this novel statistical framework get more attention. The authors mention it, but in my opinion it misses some depth and details. It seems that the simultaneous estimation is one major advantage, but it gets a bit lost in the biological interpretation of the case study (wild boar population). I think that by restructuring it to a “methodological paper”, the authors can reach a much broader audience of researchers dealing with harvest species in general.

2. This leads me directly to the next feedback point. At the moment the authors show that this “novel statistical framework” can be used for harvest species. But couldn't they expand/generalize it for species that need to be protected? The authors could also reach conservationists by showing how this method could be used for an endangered species for instance. Proper and reliable density estimation are also of huge importance for endangered species.

3. I am a bit concerned to what extent this model can be used by managers/researcher that do not have harvest records/trapping data, but only rely on camera trap data for instance. It seems that in Japan trapping records are important measure to control the population of wild boar, however, in Europe, as I know, hunting plays most important role and trapping is not allowed to control the population. That makes it difficult to translate this approach to a European situation. I would advise that the authors elaborate this issue (challenges and limitations for generalization of the statistical framework) in a separate section of the manuscript. In addition, adding information about a more general use will make it much more interesting to a broader public. I also miss this aspect in the last section of the manuscript about management implications. Can this model be used if no trapping data are available? How would I need to deal with it?

4. One confusing thing for the reader is that the authors switch between the terms “harvest record” and “trapping records”. They need to be consistent with it to not confuse the reader. In addition, it would be good to explain “harvest record” already shortly in the introduction, since it also appears in figure 1 (in the introduction). To me the difference between hunting bag and harvest record was not clear in the beginning. Defining harvest record will help the reader a lot.

5. I am a bit concerned about the fact that the authors did not include more age classes. They used only juveniles (<4 months) and adults (>4 months). As mentioned in my specific comments below, I would add yearlings because we cannot be certain that adults and yearlings do not differ in their habitat use, activity pattern and catchability.

6. I would also recommend to the authors to make clearer in the result section that the analysis were conducted on different spatial scales. At the moment it seems that some interpretations from the results are not correct, however, after reading carefully the methods again it becomes clear. As an example. In the summary, the authors mention the habitat preference of forests adjacent to abandoned fields. From the result section, this conclusion does not become really clear, still I am certain that the analysis were done on this.

7. A bit a minor remark, but since it is a general remark and not mentioned in specific minor comments: In general I would rather refer to “landscape variables” than “environmental variables”, since environmental can be much broader and are less specific. Landscape gives the reader a more concrete idea of what is meant.

You find my specific (minor) comments below. I hope they be useful

Minor comments:

Page 2:

L40: instead of “;” – sign, I would advise a point as end of the sentence.

L44: evaluated “as an example population”. I would add this to make it clearer for the reader.

L45: mainly due to harvesting

L45-L46: I do not see where this conclusion/result come from in your result section. In your results, you do not say that forests adjacent to abandoned fields are preferred and I don't think that you analysed this. If you did so, please make it clear to the reader.

L47: so throughout the season? I would add this in the sentence to make it clearer to the reader.

L50: for me it sounds that attractiveness of trap bait and activity changes in response to food scarcity is similar or at least closely related. I would rather write: ..., which probably reflected changes in the attractiveness of the trap bait due to activity changes in response to food scarcity.

Page 3:

L3: I advise putting keywords in alphabetical order.

L9: I would recommend mentioning some example mammal populations in brackets.

L16: I think I understand what you mean by field signs but maybe short elaboration is better.

Give an example for instance. I am not sure if field sign is the most appropriate word here.

L27: what is the difference between hunting bag and harvest record? I would have expected that it is the same. Thus, I would advise explaining the definition of harvest record already in an early stage since it is also used in the title.

L41: can you elaborate a bit more which parameters for instance are estimated by the camera trap data so that it is better to understand for a reader without the background knowledge of the REST model?

L51: You can even put it in a broader context. Not only in Japan but worldwide wild boar are causing damage. You could start with the fact that worldwide wild boar are causing damage with some countries as example and then narrow down to also in Japan they are causing damage and continue your story. That makes it maybe even more interesting for a broader public.

L52: “As abandonment of agricultural fields increases due to human population aging, sites preferred by wild boar, including abandoned fields and bamboo forests, have been increasing, thereby leading to crop damage near residential areas. This sentence is not really clear to me. It is a long sentence and a bit confusing. I would recommend rephrasing it.

Page 4:

L2: delete “the” in front of “environmental conditions”.

L2-L9: that paragraph is a bit confusing to me. I think the main points you want to make is that catchability, population size as well as habitat use change spatially and seasonally. Because you mention the reason for two of the factors (habitat use and population size) in between of the enumeration of the three parameters the readers gets a bit confused. I would advise rephrasing this paragraph.

L17: Why does simultaneous estimation in a Bayesian framework results in less error? I would recommend giving some additional explanation for this. Does not need to be long, but one addition sentence may help to get more clarity here.

L20: You mention here trapping records. I guess that this is the same as harvest records? But that is a guess and not really clear. I would advise using consistent words for it. And also some additional information about it. I was wondering for instance how sure you are about the fact that trapping by different municipalities and during different seasons was conducted with a constant effort? If that is an assumption you have to be quite certain about it, because I think that this can have huge impact on the results if it is not conducted with constant effort. Otherwise it might be really similar to hunting bag? I see now that you mention the trapping effort in the methods. That's good, but I would say, make sure that the reader knows here already that you take different trapping effort into account. Simply one short additional sentence should be enough.

L24-L29: I would rephrase this last part of the introduction. At the moment it feels a bit lost from the earlier paragraph. I would recommend something like: “Based on seasonal dynamics of the

three parameters, we proposed an effective trapping strategy for sites and timing of trap placement on a landscape level for an example study area in the southern part of Chiba prefecture, central Japan." By simply switching first and last part of your paragraph it better links to the earlier paragraph.

Page 5:

L1: There seems to be a clear distinction between hunting and trapping. I think that this needs to be clearer for the reader at an early stage. I start now understanding that there is a difference, but it can be much easier for the reader if it is explained clearly at the beginning of the manuscript. I think that it namely not so common to have trapping (as you define it) next to hunting methods for wild boar control. At least not in Europe. Thus, I guess that European readers may be a bit confused here.

L6-L11: I think that you can better combine those two sentences. Now it seems a bit that you rephrase the same sentence again.

L14: If I understand correctly you distinguished between juveniles (0-4 months) and adults (>4months). Wouldn't you expect differences in catchability and habitat use between piglets (<13 months), sub-adults/yearlings (13-24 months old individuals) and adults (>24 months old individuals) as well? I understand if the distinction of other age classes only on camera traps is not possible but I think that it is important to mention this possible limitation of the study here.

L21: where the camera traps randomly distributed throughout the area? I guess so because you used the Rowcliffe method, but I don't see that you mention it in this paragraph.

L35: "The environmental variable was based on land use data..." □ add "was based" to your sentence.

L37: add "average" home range? That is probably more appropriate.

L45: I think that it is clearer if you write "... six landscape elements..." instead of "six environmental elements".

L60: see my comments on the age classes before. I think some additional explanation is necessary why you did not distinguish between yearlings and adults as well. Especially when you have trapping data more than only the two age classes you distinguish may be available.

Page 6:

L9: replace "rest were" by "remaining percentage was".

L29: what is the distribution of the staying time? is there a lot of variation? I would like to see a histogram of the staying time variable in the supplementary material.

Page 8:

L17: Put posterior distributions in the supplemental material please.

L28: rather than "environmental variables", I would name it "landscape variables". That is more precise and clearer to the reader.

L39-41: I would put those two lines in the methods instead of in the results.

Page 9:

L27: "simultaneous estimation". Is that one of the major advantages of this technique/tool that you used here? If so, I would make that clear to the reader. At the moment it pops up in the title and here in the first sentence of discussion, but to me it is not really clear if that is an important point or not. Needs some clarification here. Ok, I see now in the next sentences of the discussion that you mention the advantages. I would already see those points in the introduction (when elaborating the problem).

L29: delete "quite" in sentence "quite useful". It is useful!

L40: I miss a bit more in-depth detail about how exactly this methods shows advantages. Why and how are parameter estimates more reliable?

L45: refer to the reports. How do we know what is the "correct" density? How do we know if those estimates are more reliable than others, which are also estimated? That is not explained enough here and should be more convincing.

L60: can you add and overview of number of cameras in each landscape element in the study area (in supplemental material) so that you can underline your word “mainly” by a number/percentage. That makes your sentence more convincing.

Page 10:

L4: ON average. Not “in average”.

Page 11:

L20: Can you also mention management implications for countries that do not use trapping as control measure for the population?

L39-L46: As mentioned in my general comments, you can put much more focus on this last paragraph in your introduction and make it much more interesting for a broader public.

Page 12:

L10: reference nr 17 populationabundance should be two separate words.

Page 15:

Figure 1: What are the boxes and circles indicating? Mention it in the figure description. I anyway need some addition explanation for figure 1. When you only read the introduction where the first time is referred to figure 1 and by having only those information from introduction and figure description it is difficult to understand the figure. For instance, what is meant with density/mesh? I would advise either give additional explanation in introduction and/or figure description or make an easier graphical representation for the introduction.

Page 16:

L 50: Figure 4: is juvenile density never below adult density or not visible in month December-April for the juveniles because it lies below the density of the adults? If the latter is the case I would advise to use lines instead of filling the surface. Wouldn't be nicer to add also the bars in the two different colors (light and dark grey) for the two age classes? What is the reason to use the 50% credible interval and not a 95% confidence interval or a standard error?

Page 17:

L20: put the x-axis with the month at the bottom (at $y=-1$ for instance) so that it is easier to read and not all the months names are below the lines and points. This makes the graph a bit confusing.

L54: “throughout the year” instead of “through a year”.

Supporting information:

L14: typing error: “monitoring”.

L20: Fig. S4: b) y axis is entitled with number of videos with two most left graphs, but has frequency on y-axis for the two most right and two central graphs. I think that you should remove the “frequency” on the y-axis, as it is also removed in (a). I don't understand in the figure description what you mean with y-values of circular points in the figures. I don't see any circular points.

L26: In reference Rowcliffe et al, (2014) is missing.

Reviewer: 2

Comments to the Author(s)

Dear authors,

I have found you work timely and interesting. My main concern is the potential pseudoreplication in your experimental design. I think that the incorporation of the camera id as a random term (e.g., intercept), will improve your Bayesian modelling preventing from getting spurious effects (see, Paul-Christian Bürkner. 2020. Package ‘brms’ 2.12.0 version).

Other minor comments have been included in the pdf.

I hope that my comments will be useful for your investigation
Yours sincerely

Author's Response to Decision Letter for (RSOS-200579.R0)

See Appendix B.

RSOS-200579.R1 (Revision)

Review form: Reviewer 1

Is the manuscript scientifically sound in its present form?

Yes

Are the interpretations and conclusions justified by the results?

Yes

Is the language acceptable?

Yes

Do you have any ethical concerns with this paper?

No

Have you any concerns about statistical analyses in this paper?

No

Recommendation?

Accept with minor revision (please list in comments)

Comments to the Author(s)

Dear authors,

You did a great job in improving the manuscript. I only have some minor comments on text parts that have been added in the revised version.

Minor comments:

Page 13:

Line 38: make clear that the reader understands the novelty of this approach. You say that it accounts for spatial and temporal heterogeneity in detectability. Maybe it would be good to add one sentence that in the previous approaches, this has not been done. I see that this is what you want to make say, but I would state it in an additional sentence.

Page 17:

Line 20: Put the table that you used as response to the comment from reviewer 2 in the supplement as well and refer to it here.

Page 22:

Line 27: What do you mean with "in seperation"? seperately is maybe a better word?

Line 29: You say that the density estimates become much more precise. Is it precise? Or do you mean accurate? You state it but you did not test (statistically) if the results are more precise/accurate right? I would rephrase this sentence a bit to reduce the chance of misinterpretation by the reader.

Line 30: "Of course our models can also be used for eradication...". Add also in the sentence for clarity.

Review form: Reviewer 2

Is the manuscript scientifically sound in its present form?

Yes

Are the interpretations and conclusions justified by the results?

Yes

Is the language acceptable?

Yes

Do you have any ethical concerns with this paper?

No

Have you any concerns about statistical analyses in this paper?

No

Recommendation?

Accept as is

Comments to the Author(s)

..

Decision letter (RSOS-200579.R1)

Dear Dr Miyashita:

On behalf of the Editors, I am pleased to inform you that your Manuscript RSOS-200579.R1 entitled "Simultaneous estimation of seasonal population density, habitat preference, and catchability of wild boars based on camera data and harvest records" has been accepted for publication in Royal Society Open Science subject to minor revision in accordance with the referee suggestions. Please find the referees' comments at the end of this email.

The reviewers and Subject Editor have recommended publication, but also suggest some minor revisions to your manuscript. Therefore, I invite you to respond to the comments and revise your manuscript.

- Ethics statement

- Data accessibility

<http://datadryad.org/submit?journalID=RSOS&manu=RSOS-200579.R1>

- Competing interests

- Authors' contributions

- Acknowledgements

- Funding statement

Because the schedule for publication is very tight, it is a condition of publication that you submit the revised version of your manuscript before 22-Jul-2020. Please note that the revision deadline

will expire at 00.00am on this date. If you do not think you will be able to meet this date please let me know immediately.

Reviewer comments to Author:
Reviewer: 2

Comments to the Author(s)

..

Reviewer: 1

Comments to the Author(s)

Dear authors,

You did a great job in improving the manuscript. I only have some minor comments on text parts that have been added in the revised version.

Minor comments:

Page 13:

Line 38: make clear that the reader understands the novelty of this approach. You say that it accounts for spatial and temporal heterogeneity in detectability. Maybe it would be good to add one sentence that in the previous approaches, this has not been done. I see that this is what you want to make say, but I would state it in an additional sentence.

Page 17:

Line 20: Put the table that you used as response to the comment from reviewer 2 in the supplement as well and refer to it here.

Page 22:

Line 27: What do you mean with "in seperation"? seperately is maybe a better word?

Line 29: You say that the density estimates become much more precise. Is it precise? Or do you mean accurate? You state it but you did not test (statistically) if the results are more precise/accurate right? I would rephrase this sentence a bit to reduce the chance of misinterpretation by the reader.

Line 30: "Off course our models can also be used for eradication...". Add also in the sentence for clarity.

Author's Response to Decision Letter for (RSOS-200579.R1)

See Appendix C.

Decision letter (RSOS-200579.R2)

Dear Dr Miyashita,

It is a pleasure to accept your manuscript entitled "Simultaneous estimation of seasonal population density, habitat preference, and catchability of wild boars based on camera data and harvest records" in its current form for publication in Royal Society Open Science.

Appendix A**ROYAL SOCIETY
OPEN SCIENCE****Simultaneous estimation of seasonal population density,
habitat preference, and catchability of wild boars based on
camera data and harvest records**

Journal:	Royal Society Open Science
Manuscript ID	RSOS-200579
Article Type:	Research
Date Submitted by the Author:	05-Apr-2020
Complete List of Authors:	Yokoyama, Yuichi; University of Tokyo, Graduate School of Agriculture and Life Sciences Nakashima, Yoshihiro; Nihon University, College of Bioresource Science Yajima, Gota; Nihon University, College of Bioresource Science Miyashita, Tadashi; University of Tokyo, Agriculture and Life Sciences
Subject:	ecology < BIOLOGY, environmental science < BIOLOGY
Keywords:	wildlife management, population dynamics, habitat selection
Subject Category:	Ecology, Conservation, and Global Change Biology

Author-supplied statements

Relevant information will appear here if provided.

Ethics

Does your article include research that required ethical approval or permits?:

This article does not present research with ethical considerations

Statement (if applicable):

Harvesting records used in this study were provided by local governments in Chiba prefecture, and the harvesting was conducted by the local governments under Wildlife Protection and Hunting Management Law in Japan.

Data

It is a condition of publication that data, code and materials supporting your paper are made publicly available. Does your paper present new data?:

Yes

Statement (if applicable):

The datasets supporting this article have been uploaded as part of the supplementary material. The details are, 1) R code, 2) readme file, 3) adult boar file, 4) juvenile boar file, 5) environmental file.

Conflict of interest

I/We declare we have no competing interests

Statement (if applicable):

CUST_STATE_CONFLICT :No data available.

Authors' contributions

This paper has multiple authors and our individual contributions were as below

Statement (if applicable):

YY carried out data analysis, drafted the manuscript; YN participated in the design of the study, collected field data, and helped data analysis; GY collected field data and analysed video recordings; TM participated in the design of the study and critically revised the manuscript. All authors gave final approval for publication and agree to be held accountable for the work performed therein.

Simultaneous estimation of seasonal population density, habitat preference, and catchability of wild boars based on camera data and harvest records

Yuichi Yokoyama¹, Yoshihiro Nakashima², Gota Yajima², Tadashi Miyashita¹

1 Graduate School of Agriculture and Life Sciences, University of Tokyo, Bunkyo Ward, Tokyo, 113-8657, Japan

2 College of Bioresource Science, Nihon University, Fujisawa, Kanagawa, 252-0880, Japan

Keywords: population dynamics; random encounter and staying time model; seasonal habitat preference; *Sus scrofa*; ecosystem management

1. Summary

Analyses of life history and population dynamics are essential for effective population control of wild mammals. We developed a model for the simultaneous estimation of seasonal changes in three parameters—population density, habitat preference, and trap catchability of target animals—based on camera-trapping data and harvest records. The random encounter and staying time model, with no need to individual recognition, is the core component of the model; by combining this model with harvest records, we estimated density at broad spatial scales and catchability by traps. Here, the wild boar population in central Japan was evaluated. We found that the estimated population density increased after the birth period and then decreased until the next birth period, due mainly to harvesting. Habitat preference changed seasonally, but forests adjacent to abandoned fields were generally preferred. These patterns can be explained by patterns of food availability and resting or nesting sites. Catchability by traps also changed seasonally, with relatively high values in the winter, which probably reflected changes in the attractiveness of the trap bait or activity changes in response to food scarcity. Based on these results, we proposed an effective trapping strategy for wild boars, focusing on when and where to set traps.

*Author for correspondence: Tadashi Miyashita (tmiya@es.a.u-tokyo.ac.jp).

†Present address: Graduate School of Agriculture and Life Sciences, University of Tokyo, Bunkyo Ward, Tokyo, 113-8657, Japan

Keywords

population dynamics; wildlife management; seasonal habitat preference; *Sus scrofa*; ecosystem management

6 2. Introduction 7

Excessive increases in some mammal populations have a wide range of negative consequences, such as
degradation of ecosystems [1], agricultural damage [2,3], and **disease transmission [4]**. Implementation of
effective management strategies for wildlife populations requires detailed knowledge on the life history and
population dynamics of the target species [5-8]. However, mammals in the wild are often cryptic with rare
sightings; even their field signs are often difficult to find.

The wild boar (*Sus scrofa*), with its broad geographical range, is one such species [9,10]. It causes crop damage
[11-13] and spreads disease to livestock [14,15]. Although population density and seasonal dynamics are key
factors for the effective management of wild boar [16], conventional methods for estimating their numbers are
not sufficiently precise or accurate and therefore a standard methodological framework is lacking [17].
Estimating wild boar population sizes by direct observation is difficult due to the preference for dense
understory shrubs and nocturnal and cautious behaviors [18,19]. Indirect indices, such as hunting bag
statistics or pellet counts, have been used to estimate population density (e.g., 20-22), but these indices can be
affected by several factors, including season, weather conditions, and visibility in habitats [17]. **They are thus**
**context-dependent and not reliable due to seasonal and spatial variation.**

Camera traps represent a potential alternative for the reliable estimation of the absolute density of wild boars.
Recently developed approaches to estimate animal density without individual recognition might be applicable
to wild boars. Rowcliffe *et al.* [23] presented the random encounter model (REM) based on ideal gas models
[24]. Nakashima, Fukasawa & Samejima [25] improved the feasibility of REM by developing the random
encounter and staying time (REST) model, in which all required parameters are estimated exclusively by
camera trap data. More recently, Nakashima, Hongo & Akomo-Okoue [26] have incorporated habitat
covariates into the REST model, allowing for likelihood-based estimation of the relationship between habitats
and animal density. This approach yields reliable density estimates at the landscape scale, accounting for
spatial and temporal heterogeneity in animal detectability.

In Japan, the spread of wild boar populations has caused serious agricultural damage [27]. Wild boars
accounted for about 30% of total crop damage caused by wildlife in 2018 [28]. As abandonment of agricultural
fields increases due to human population aging, sites preferred by wild boar, including abandoned fields and
bamboo forests, have been increasing, thereby leading to crop damage near residential areas [29]. Box and
snare traps are typically used to control wild boar populations [30], but trapping is based on empirical or
anecdotal knowledge, as opposed to an informed and systematically designed approach. Moreover, the
hunter population is aging and decreasing [31,32], emphasizing the need for higher trap catchability with less

1
2 effort. Catchability is affected by the environmental conditions and season [33,34]; accordingly, seasonal and
3
4 spatial variation in catchability need to be clarified at the landscape level. Furthermore, 
[revised manuscript text omitted]

where i indicates the camera ID ($i = 1, 2, \dots, 180$) and m indicates the month ($m = 1, 2, \dots, 12$). Because wild boar activity is assumed to change seasonally, the staying time within the focal area was measured for each month. About 50 videos were selected from several sites each month, and staying time was measured using a stopwatch in a laboratory. To estimate the average staying time for each month, staying time data were fit to four distributions (exponential, gamma, log-normal, and Weibull distributions), and the best-fit model was determined by the Wyckoff -Akaike information criterion (WAIC) [46]. The log-normal distribution fitted the data the best. The staying time of juveniles and adults were estimated separately. The density of juveniles was estimated only from June to October in 2018 and in May in 2019 because the counts were too small to estimate staying time in other months. The "active time" in equation (1) was calculated from the total recording period

multiplied by the “daily activity proportion of time.” The daily activity proportion of time was estimated as the daily change in the number of times wild boars were recorded by camera [47], or the proportion of active time in a 24-h period. The REST model assumes that all individuals are active at the peak of the activity rhythm [25]. Because the daily activity proportion may differ among months and ages, separate estimates were obtained for juveniles and adults each month (see supplementary material figure S4). The negative binomial distribution was fitted to the number of passages through the focal area, as in Nakashima *et al.* [25]. The details of the REST model and analysis are described in Nakashima *et al.* [25].

Seasonal habitat use was assessed by incorporating environmental variables (summarized as PCA axes) as covariates into the REST model, as expressed in equation (2). The coefficient α associated with environmental variables was used as the index of habitat use in the home range, as α indicates fine-scale spatio-temporal variation in wild boar densities. The following equation was used to assess habitat use:

$$[\text{Density}]_{i,m} = \exp(\alpha_{0,m} + \sum_{e=1}^4 \alpha_{e,m} \times [\text{PC}]_{e,i}), \text{ eqn (2)}$$

where i indicates the camera ID ($s = 1, 2, \dots, 180$), m indicates the month ($m = 1, 2, \dots, 12$), and e indicates the type of environmental variable. PC indicates the principal component scores summarizing environmental variables around each camera, α_0 represents for the intercept (average density across environments), and α_1 to α_4 represent how each landscape composition affects the local wild boar density (i.e., the habitat preference in each month).

(2) Population density at the management-unit level

To estimate the wild boar density at the management-unit level, densities in 1 km² grid cells over the whole study area were estimated by equation (3), assuming that landscape variables determine density:

$$[\text{Density}]_{c,m} = \exp(\alpha_{0,m} + \sum_{e=1}^4 \alpha_{e,m} \times [\text{PC}]_{e,c}), \text{ eqn (3)}$$

[revised manuscript text omitted]

One of the advantages of camera-trapping observations is that all animals passing in front of cameras are
detectable. We focused on the wild boar, but the methodology can be readily applied to sympatric species.
Moreover, camera-trapping data spanning several years make it possible to estimate demographic parameters
(such as survival rates) and long-term population dynamics. These technical advances will make a substantial
contribution to the management of wildlife that are difficult to detect by traditional methods.

**Acknowledgments**

We thank the public officers of the municipal governments in our study area. We are also grateful to H. Yokomizo, G.
Takimoto and G. Fujita for helpful comments. Data collection was supported by students of Nihon University College of
Bioresource Science. We appreciate their huge efforts.

References

1. Côté S.D., Rooney T.P., Tremblay J.P., Dussault C., & Waller D.M.. 2004. Ecological impacts of deer overabundance. *Annu. Rev. Ecol. Evol. Syst.* **35**, 113-147. (doi: [10.1146/annurev.ecolsys.35.021103.105725](https://doi.org/10.1146/annurev.ecolsys.35.021103.105725))
2. Takatsuki S.. 2009. Effects of sika deer on vegetation in Japan: a review. *Biol Conserv.* **142(9)**. 1922-1929. (doi: [10.1016/j.biocon.2009.02.011](https://doi.org/10.1016/j.biocon.2009.02.011))
3. Ward A.I.. 2005. Expanding ranges of wild and feral deer in Great Britain. *Mammal Rev.* **35(2)**. 165-173. (doi: [10.1111/j.1365-2907.2005.00060.x](https://doi.org/10.1111/j.1365-2907.2005.00060.x))
4. Ward A.I., & Smith G.C.. 2012. Predicting the status of wild deer as hosts of *Mycobacterium bovis* infection in Britain. *Eur J Wildlife Res.* **58(1)**. 127-135. (doi: [10.1007/s10344-011-0553-7](https://doi.org/10.1007/s10344-011-0553-7))
5. Apollonio, M., Belkin, V. V., Borkowski, J., Borodin, O. I., Borowik, T., Cagnacci, F., Danilkin, A. A., Danilov, P. I., Faybich, A., Ferretti, F., Gaillard, J. M., Hayward, M., Heshtaut, P., Heurich, M., Hurynovich, A., Kashtalyan, A., Kerley, G. I. H., Kjellander, P., Kowalczyk, R., Kozorez, A., Matveytchuk, S., Milner, J. M., Mysterud, A., Ozoliņš, J., Panchenko, D.V., Peters, W., Podgórski, W. P. T., Pokorny, B., Rolandsen, C. M., Ruusila, V., Schmidt, K., Sipko, T. P., Veeroja, R., Velihurau, P., & Yanuta, G.. 2017. Challenges and science-based implications for modern management and conservation of European ungulate populations. *Mammal Res.* **62(3)**. 209-217. (doi: [10.1007/s13364-017-0321-5](https://doi.org/10.1007/s13364-017-0321-5))
6. Beasley, J. C., Ditchkoff, S. S., Mayer, J. J., Smith, M. D., & Vercauteren, K. C.. 2018. Research priorities for managing invasive wild pigs in North America. *J Wildlife Manage.* **82(4)**. 674-681. (doi: [10.1201/b22014-20](https://doi.org/10.1201/b22014-20))
7. Grarock, K., Tidemann, C. R., Wood, J. T., & Lindenmayer, D. B.. 2014. Understanding basic species population dynamics for effective control: a case study on community-led culling of the common myna (*Acridotheres tristis*). *Biol Invasions.* **16(7)**. 1427-1440. (doi: [10.1007/s10530-013-0580-2](https://doi.org/10.1007/s10530-013-0580-2))
8. Hulme, P. E.. 2006. Beyond control: wider implications for the management of biological invasions. *J Appl Ecol.* **43(5)**. 835-847. (doi: doi.org/10.1111/j.1365-2664.2006.01227.x)
9. Barrios-García, M. N., & Ballari, S. A.. 2012. Impact of wild boar (*Sus scrofa*) in its introduced and native range: a review. *Biol Invasions.* **14(11)**, 2283-2300. (doi: [10.1007/s10530-012-0229-6](https://doi.org/10.1007/s10530-012-0229-6))
10. Massei, G., & Genov, P. V.. 2004. The environmental impact of wild boar. *Galemys.* **16(1)**. 135-145. (doi:)
11. Ballari, S. A., & Barrios-García, M. N.. 2014. A review of wild boar *Sus scrofa* diet and factors affecting food selection in native and introduced ranges. *Mammal Rev.* **44(2)**. (doi: [10.1111/mam.12015](https://doi.org/10.1111/mam.12015))
12. Herrero, J., García-Serrano, A., Couto, S., Ortuño, V. M., & García-González, R.. 2006. Diet of wild boar *Sus scrofa* L. and crop damage in an intensive agroecosystem. *Eur J Wildlife Res.* **52(4)**. 245-250. (doi: [10.1007/s10344-006-0045-3](https://doi.org/10.1007/s10344-006-0045-3))
13. Pimentel, D., Zuniga, R., & Morrison, D.. 2005. Update on the environmental and economic costs associated with alien-invasive species in the United States. *Ecol Econ.* **52(3)**. 273-288. (doi: [10.1016/j.ecolecon.2004.10.002](https://doi.org/10.1016/j.ecolecon.2004.10.002))
14. Barasona, J. A., Latham, M. C., Acevedo, P., Armenteros, J. A., Latham, A. D. M., Gortazar, C., Carro, F., Soriguer, R. C., & Vicente, J.. 2014. Spatiotemporal interactions between wild boar and cattle: implications for cross-species disease transmission. *Vet Res.* **45(1)**. 122. (doi: [10.1186/s13567-014-0122-7](https://doi.org/10.1186/s13567-014-0122-7))
15. Naranjo, V., Gortazar, C., Vicente, J., & de la Fuente, J.. 2008. Evidence of the role of European wild boar as a reservoir of *Mycobacterium tuberculosis* complex. *Vet Microbiol.* **127(1-2)**. 1-9. (doi: doi.org/10.1016/j.vetmic.2007.10.002)
16. Massei, G., Coats, J., Lambert, M. S., Pietravalle, S., Gill, R., & Cowan, D.. 2018. Camera traps and activity signs to estimate wild boar density and derive abundance indices. *Pest Manag Sci.* **74(4)**. 853-860. (doi: doi.org/10.1002/ps.4763)
17. Keuling, O., Sange, M., Acevedo, P., Podgórski, T., Smith, G., Scandura, M., Apollonio, M., Ferroglio, E., & Vicente, J.. 2018. Guidance on estimation of wild boar population abundance and density: methods, challenges, possibilities. *EFSA Support Publ.* **15(6)**. (doi: [10.2903/sp.efsa.2018.EN-1449](https://doi.org/10.2903/sp.efsa.2018.EN-1449))
18. Ohashi, H., Saito, M., Horie, R., Tsunoda, H., Noba, H., Ishii, H., Kuwabara, T., Hiroshige, Y., Koike, S., Hoshino, Y., Toda, H., & Kajii, K.. 2013. Differences in the activity pattern of the wild boar *Sus scrofa* related to human disturbance. *Eur J Wildlife Res.* **59(2)**. 167-177. (doi: [10.1007/s10344-012-0661-z](https://doi.org/10.1007/s10344-012-0661-z))
19. Sweitzer, R. A., Van Vuren, D., Gardner, I. A., Boyce, W. M., & Waithman, J. D.. 2000. Estimating sizes of wild pig populations in the north and central coast regions of California. *J Wildlife Manage.* 531-543. (doi: [10.2307/3803251](https://doi.org/10.2307/3803251))
20. Acevedo, P., Vicente, J., Höfle, U., Cassinello, J., RUIZ-FONS, F., & Gortázar, C.. 2007. Estimation of European wild boar relative abundance and aggregation: a novel method in epidemiological risk assessment. *Epidemiol Infect.* **135(3)**. 519-527. (doi: [10.1017/S0950268806007059](https://doi.org/10.1017/S0950268806007059))
21. Davis, A.J., Hooten, M.B., Miller, R.S., Farnsworth, M.L., Lewis, J., Moxcey, M. and Pepin, K.M.. 2016. Inferring invasive species abundance using removal data from management actions. *Ecol Appl.* **26**. 2339-2346. (doi: [10.1002/eap.1383](https://doi.org/10.1002/eap.1383))
22. Osada, Y., Kuriyama, T., Asada, M., Yokomizo, H., & Miyashita, T.. 2015. Exploring the drivers of wildlife population dynamics from insufficient data by Bayesian model averaging. *Popul Ecol.* **57(3)**. 485-493. (doi: [10.1007/s10144-015-0498-x](https://doi.org/10.1007/s10144-015-0498-x))
23. Rowcliffe, J. M., Field, J., Turvey, S. T., & Carbone, C. 2008. Estimating animal density using camera traps without the need for individual recognition. *J Appl Ecol.* **45(4)**. 1228-1236. (doi: [10.1111/j.1365-2664.2008.01473.x](https://doi.org/10.1111/j.1365-2664.2008.01473.x))
24. Hutchinson, J. M., & Waser, P. M.. 2007. Use, misuse and extensions of "ideal gas" models of animal encounter. *Biol Rev.* **82(3)**.

- 335-359. (doi: doi.org/10.1111/j.1469-185X.2007.00014.x)
25. Nakashima, Y., Fukasawa, K., & Samejima, H.. 2018. Estimating animal density without individual recognition using information derivable exclusively from camera traps. *J Appl Ecol.* **55**(2). 735-744. (doi: [10.1111/1365-2664.13059](https://doi.org/10.1111/1365-2664.13059))
 26. Nakashima, Y., Hongo, S., & Akomo-Okoue, E. F.. 2020. Landscape-scale estimation of forest ungulate density and biomass using camera traps: Applying the REST model. *Biol Conserv.* **241**. 108381. (doi: [10.1016/j.biocon.2019.108381](https://doi.org/10.1016/j.biocon.2019.108381))
 27. Kuwabara T., Ohashi H., Saito M., Hiroshige Y., Koike S., Toda H., & Kaji K.. 2010. Conditions and course of solution on rural agricultural damages caused by wild boar. *J Rural Econ Special Issue.* **2010**. 305 – 312 (in Japanese)
 28. Ministry of Agriculture, Forestry and Fisheries in Japan. 2019. Annual reports on the reality and countermeasure of wildlife damage. URL <https://www.maff.go.jp/j/seisan/tyozyu/higai/attach/pdf/index-326.pdf> (in Japanese)
 29. Kodera Y., Kanzaki N., Kaneko Y., & Tokida K. 2001.. Habitat selection of Japanese wild boar in Iwami district, Shimane prefecture, western Japan. *Wildl Conserv Jpn.* **6**. 119–129. (doi: [10.20798/wildlifeconsjp.6.2_119](https://doi.org/10.20798/wildlifeconsjp.6.2_119)) (in Japanese)
 30. Saito, M., Momose, H., & Mihira, T.. 2011. Both environmental factors and countermeasures affect wild boar damage to rice paddies in Boso Peninsula, Japan. *Crop Prot.* **30**(8). 1048-1054. (doi: [10.1016/j.cropro.2011.02.017](https://doi.org/10.1016/j.cropro.2011.02.017))
 31. Eguchi, Y.. 2008. Crop damage management: management of damage by wild boar. In: Takatsuki, S., Yamagiwa, J. (Eds.), Middle- and Large-Sized Mammals Including Primates (Mammalogy in Japan 2). University of Tokyo Press, Tokyo. pp. 401-426 (in Japanese).
 32. Ueda, G., Kanzaki, N., & Koganezawa, M. (2010). Changes in the structure of the Japanese hunter population from 1965 to 2005. *Human Dimensions of wildlife.* **15**(1). 16-26. (doi: [10.1080/10871200903161470](https://doi.org/10.1080/10871200903161470))
 33. Caley, P.. 1994. Factors affecting the success rate of traps for catching feral pigs in a tropical habitat. *Wildlife Res.* **21**(3). 287-291. (doi: [10.1071/WR9940287](https://doi.org/10.1071/WR9940287))
 34. Wyckoff, A. C., Henke, S. E., Campbell, T., & VerCauteren, K. C.. 2006. Is trapping success of feral hogs dependent upon weather conditions?. In Proceedings of the Vertebrate Pest Conference (Vol. 22, No. 22). (doi: [10.5070/V422110217](https://doi.org/10.5070/V422110217))
 35. Morelle, K., Podgórski, T., Prévot, C., Keuling, O., Lehaire, F., & Lejeune, P. 2015. Towards understanding wild boar *Sus scrofa* movement: a synthetic movement ecology approach. *Mammal Rev.* **45**(1). 15-29. (doi: [10.1111/mam.12028](https://doi.org/10.1111/mam.12028))
 36. Rastetter, E. B., King, A. W., Cosby, B. J., Hornberger, G. M., O'Neill, R. V., & Hobbie, J. E.. 1992. Aggregating fine - scale ecological knowledge to model coarser - scale attributes of ecosystems. *Ecol Appl.* **2**(1). 55-70. (doi: [10.2307/1941889](https://doi.org/10.2307/1941889))
 37. Asada, M.. 2011 Distribution and pest control, damage to agricultural reduction for wild boar in 2009 in Chiba prefecture, Japan. *Rep Chiba Biodivers Cent.* **3**. 49–64 (in Japanese)
 38. Ministry of the Environment in Japan. 2009. The 6th and 7th national survey on the natural environment. URL http://www.biodic.go.jp/kiso/fnd_f.html (in Japanese)
 39. Suzuki, M., Miyashita, T., Kabaya, H., Ochiai, K., Asada, M., & Tange, T.. 2008. Deer density affects ground - layer vegetation differently in conifer plantations and hardwood forests on the Boso Peninsula, Japan. *Ecol Res.* **23**(1). 151-158. (doi: [10.1007/s11284-007-0348-1](https://doi.org/10.1007/s11284-007-0348-1))
 40. Chiba Prefectural Government (2018). Response to wildlife damage: Yearly report of crop damage. URL https://www.pref.chiba.lg.jp/noushin/chouju/yuugai/documents/h30higaijyoukyou23_4.pdf (in Japanese)
 41. Tsuji, T., Yokoyama, M., Asano, M., & Suzuki, M.. 2013. Estimation of the fertility rates of Japanese wild boars (*Sus scrofa leucomystax*) using fetuses and corpora albicans. *Acta Theriol.* **58**(3). 315-323. (doi: [10.1007/s13364-012-0115-8](https://doi.org/10.1007/s13364-012-0115-8))
 42. Thurfjell, H., Ball, J. P., Åhlén, P. A., Kornacher, P., Dettki, H., & Sjöberg, K.. 2009. Habitat use and spatial patterns of wild boar *Sus scrofa* (L.): agricultural fields and edges. *Eur J Wildlife Res.* **55**(5). 517-523. (doi: [10.1007/s10344-009-0268-1](https://doi.org/10.1007/s10344-009-0268-1))
 43. Tsunoda, Y., Ohashi, H., Saito, M., Horie, R., Noba., H, Koike, S., Hoshino, Y., Toda, H., & Kaji, K.. 2014. Foraging condition of wild boars in Shingo district and Himuro district, Sano city, Tochigi prefecture, Japan. *Wildlife and Human Society.* **1**(2). 61-70. (doi: [10.20798/awhswhs.1.2_61](https://doi.org/10.20798/awhswhs.1.2_61)) (in Japanese)
 44. Dardaillon, M.. 1988. Wild boar social groupings and their seasonal changes in the Camargue, southern France. *Zeitschrift für Säugetierkunde,* **53**(1). 22-30.
 45. Kodera Y, Nagatsuma T, Sawada S, Fujihara S., & Kanamori H. 2010. How does spreading maize on fields influence behavior of wild boars (*Sus scrofa*)? *Mamm Sci.* **50**. 137 – 144. (doi: [10.11238/mammalianscience.50.137](https://doi.org/10.11238/mammalianscience.50.137)) (in Japanese with English abstract)
 46. Watanabe, S.. 2013. A widely applicable Bayesian information criterion. *J Mach Learn Res,* **14**(Mar). 867-897.
 47. Rowcliffe, J. M., Kays, R., Kranstauber, B., Carbone, C., & Jansen, P. A.. 2014. Quantifying levels of animal activity using camera trap data. *Methods Ecol Evol.* **5**(11). 1170-1179. (doi: doi.org/10.1111/2041-210X.12278)
 48. Gelman, A., Carlin, J. B., Stern, H. S., Dunson, D. B., Vehtari, A., & Rubin, D. B.. 2013. Bayesian data analysis. 3rd edn. Chapman and Hall/CRC. (doi: [10.1201/b16018](https://doi.org/10.1201/b16018))
 49. Plummer, M.. 2003. JAGS: A program for analysis of Bayesian graphical models using Gibbs sampling. In Proceedings of the 3rd international workshop on distributed statistical computing (Vol. 124, No. 125, p. 10). URL <https://www.r-project.org/conferences/DSC-2003/Proceedings/Plummer.pdf>
 50. R Core Team (2018). R: A language and environment for statistical computing. R Foundation for Statistical Computing, Vienna, Austria. URL <https://www.R-project.org/>
 51. Russo, L., Massei, G., & Genov, P. V.. 1997. Daily home range and activity of wild boar in a Mediterranean area free from hunting. *Ethol Ecol Evol,* **9**(3). 287-294. (doi: [10.1080/08927014.1997.9522888](https://doi.org/10.1080/08927014.1997.9522888))
 52. Fonseca, C.. 2008. Winter habitat selection by wild boar *Sus scrofa* in southeastern Poland. *Eur J Wildlife Res.* **54**(2). 361. (doi: [10.1007/s10344-007-0144-9](https://doi.org/10.1007/s10344-007-0144-9))

53. Herrero, J., Irizar, I., Laskurain, N. A., García - Serrano, A., & García - González, R.. 2005. Fruits and roots: wild boar foods during the cold season in the southwestern Pyrenees. *Ital J Zool*, **72(1)**, 49-52. (doi: [10.1080/11250000509356652](https://doi.org/10.1080/11250000509356652))
54. Kato, H., & Fukuyama, K.. 2015. Seasonal change of larger soil fauna in aging cedar forests in Boso Peninsula, Chiba prefecture. *Edaphologia*. **96**. 13-18. (doi: [10.20695/edaphologia.96.0_13](https://doi.org/10.20695/edaphologia.96.0_13)) (in Japanese)
55. Chiba Forestry Research Institute (2018). A need to take steps on bamboo forests to decrease crop damage caused by wild boars. URL https://www.pref.chiba.lg.jp/lab-nourin/seikanokouhyou/h29/documents/18-2_3.pdf (in Japanese)
56. Kodera Y., & Kanzaki N.. 2001. Seasonal change of diet and nutritional level of wild boar in Iwami district, Shimane prefecture, western Japan. *Wildl Conserv Jpn*. **6**. 109–117. (doi: [10.20798/wildlifeconsjp.6.2_109](https://doi.org/10.20798/wildlifeconsjp.6.2_109)) (in Japanese)
57. Thurfjell, H., Spong, G., & Ericsson, G..2013. Effects of hunting on wild boar *Sus scrofa* behaviour. *Wildlife Biol*. **19(1)**. 87-94. (doi: [10.2981/12-027](https://doi.org/10.2981/12-027))
58. Massei, G., Genov, P. V., & Staines, B. W.. 1996. Diet, food availability and reproduction of wild boar in a Mediterranean coastal area. *Acta Theriol*. **41**. 307-320. (doi: [10.4098/AT.arch.96-29](https://doi.org/10.4098/AT.arch.96-29))
59. Schley, L., & Roper, T. J.. 2003. Diet of wild boar *Sus scrofa* in Western Europe, with particular reference to consumption of agricultural crops. *Mammal rev*, **33(1)**, 43-56. (doi: [10.1046/j.1365-2907.2003.00010.x](https://doi.org/10.1046/j.1365-2907.2003.00010.x))

Ethical Statement

Harvesting records used in this study were provided by local governments in Chiba prefecture, and the harvesting was conducted by the local governments under Wildlife Protection and Hunting Management Law in Japan.

Funding Statement

This study was financially supported by the Grant-in-Aid for JSPS Fellows (KAKENHI Number 17H01916) and the Environmental Technology Development Fund, Ministry for the Environment, Japan (SUSHINHI Number 4-1704).

Data Accessibility

The datasets supporting this article is available in the electronic supplementary material.

Competing Interests

We have no competing interest.

Authors' Contributions

YY carried out data analysis, drafted the manuscript; YN participated in the design of the study, collected field data, and helped data analysis; GY collected field data and analysed video recordings; TM participated in the design of the study and critically revised the manuscript. All authors gave final approval for publication and agree to be held accountable for the work performed therein.

Figure 1

Figure 2

Figure 3

Figure 4

Figure 5

Figure 6

Figure captions

Figure 1 General framework of our statistical model.

Parameters we estimated from the equations of our model are shown in bold font. We assessed the seasonal change of these parameters through a year.

Figure 2 The study area consisting of 33 management units of wild boars (areas enclosed with lines) and the locations where camera traps were placed (dots).

Figure 3 Factor loadings of each landscape element that constitute Principal Component axes.

Figure 4 Seasonal changes in the estimated densities of juveniles (gray line) and adults (black line) . Whiskers
represent the 50% credible interval.

Figure 5 Estimated values of coefficients associated with the environmental variables. Filled symbols indicate
coefficients whose 95 % credible intervals do not overlap 0.

Figure 6 Seasonal change in estimated catchability of two types of traps (box and snare traps) in each month.
Whiskers represent the 50% credible interval.

Appendix B

Responses to Reviewer: 1

Thank you for valuable comments. Most of them are very useful for improving our manuscript, and we have revised it accordingly. **The revised parts are indicated by red letters in the revised manuscript.** Below you can see the responses to each of your comments.

1. I think that this manuscript is written as a standard research paper right now but has the potential to be written up as a more methodological paper. At the moment the authors mention that they introduce a novel statistical framework, but as reader you do not get the feeling that this is the main focus. I would advise restructuring the manuscript and put the main focus on the method development and using the wild boar population as a case study. Annotations in the attached r-script can help to make it reproducible for researchers dealing with similar topics/challenges. That will also involve that the advantages (“simultaneous estimation”) of this novel statistical framework get more attention. The authors mention it, but in my opinion it misses some depth and details. It seems that the simultaneous estimation is one major advantage, but it gets a bit lost in the biological interpretation of the case study (wild boar population). I think that by restructuring it to a “methodological paper”, the authors can reach a much broader audience of researchers dealing with harvest species in general.

→ We had an intention that this is a methodology-oriented paper, rather than focusing on wild boars. Nonetheless, as you pointed out, readers may feel this paper to be case study oriented. We have rewritten the Introduction so that more general issues are addressed. In particular, the order of paragraphs has been changed (L23-36). The two responses (revisions) to your comments below may also raise the impression that this paper is methodology oriented, rather than species specific.

2. This leads me directly to the next feedback point. At the moment the authors show that this “novel statistical framework” can be used for harvest species. But couldn't they expand/generalize it for species that need to be protected? The authors could also reach conservationists by showing how this method could be used for an endangered species for instance. Proper and reliable density estimation are also of huge importance for endangered species.

→ We agree with you, and thank you for helpful advice. Our method can be applied to various situations, including endangered species conservation and invasive alien management. For instance, wildlife populations in tropics are often hunted by local people to get bushmeat, and hunting records are available. Combined with a high resolution camera data, it is possible to know precise estimation of population densities, which is useful for making balance between conservation and utilization. Of course our models can be used for eradication programs of invasive aliens. These statements have been added to the Abstract (L13-14), Introduction (L68-70), and Discussion (L313–323).

3. I am a bit concerned to what extent this model can be used by managers/researcher that do not have harvest records/trapping data, but only rely on camera trap data for instance. It seems that in Japan trapping records are important measure to control the population of wild boar, however, in Europe, as I know, hunting plays most important role and trapping is not allowed to control the population. That makes it difficult to

translate this approach to a European situation. I would advise that the authors elaborate this issue (challenges and limitations for generalization of the statistical framework) in a separate section of the manuscript. In addition, adding information about a more general use will make it much more interesting to a broader public. I also miss this aspect in the last section of the manuscript about management implications. Can this model be used if no trapping data are available? How would I need to deal with it?

→ Our model used trapping data indeed, but the essence of our model is to integrate “catch-effort model” and REST model (using camera data), so it can be readily used for any types of hunting data, including ground and aerial shooting, if effort data are available. This means that our model is quite general for wildlife management. This sort of statement has been added to the Discussion, along with the model’s utility for alien species management and endangered species conservation mentioned above (L313–323).

4. One confusing thing for the reader is that the authors switch between the terms “harvest record” and “trapping records”. They need to be consistent with it to not confuse the reader. In addition, it would be good to explain “harvest record” already shortly in the introduction, since it also appears in figure 1 (in the introduction). To me the difference between hunting bag and harvest record was not clear in the beginning. Defining harvest record will help the reader a lot.

→ As you suggested, we have made the usage of terms being consistent. The harvest record we used in this study is only trapping data, as the number of shot individuals accounted for only 3% of the total number of hunted. Such a definition has been described in M & M.

5. I am a bit concerned about the fact that the authors did not include more age classes. They used only juveniles (<4 months) and adults (>4 months). As mentioned in my specific comments below, I would add yearlings because we cannot be certain that adults and yearlings do not differ in their habitat use, activity pattern and catchability.

→ From camera observation, yearlings without stripe patterns are indistinguishable from adults, so only the two age groups (juvenile and adult) were used in this study. This statement has been added (L95-97).

6. I would also recommend to the authors to make clearer in the result section that the analysis were conducted on different spatial scales. At the moment it seems that some interpretations from the results are not correct, however, after reading carefully the methods again it becomes clear. As an example. In the summary, the authors mention the habitat preference of forests adjacent to abandoned fields. From the result section, this conclusion does not become really clear, still I am certain that the analysis were done on this.

→ As you already noticed, habitat preference of wild boars was estimated from surrounding landscape structures that are spatially broader than the local camera sites. We have rephrased to make the meaning clearer, “landscape variables surrounding camera sites set in forests” (L205), and “landscape variables around camera traps”(L219) in the results section. The corresponding part of the abstract has also been changed; “forests having abandoned fields nearby were generally preferred”(L9).

7. A bit a minor remark, but since it is a general remark and not mentioned in specific minor comments: In general I would rather refer to “landscape variables” than “environmental variables”, since environmental can be much broader and are less specific. Landscape gives the reader a more concrete idea of what is meant.

→ Done.

Minor comments:

Page 2:

L40: instead of “;” – sign, I would advise a point as end of the sentence.

Done

L44: evaluated “as an example population”. I would add this to make it clearer for the reader.

Done

L45: mainly due to harvesting

Done

L45-L46: I do not see where this conclusion/result come from in your result section. In your results, you do not say that forests adjacent to abandoned fields are preferred and I don't think that you analysed this. If you did so, please make it clear to the reader.

→As with the above response to the major comment 6, we have rephrased to become the meaning clearer.

L47: so throughout the season? I would add this in the sentence to make it clearer to the reader.

Done

L50: for me it sounds that attractiveness of trap bait and activity changes in response to food scarcity is similar or at least closely related. I would rather write:, which probably reflected changes in the attractiveness of the trap bait due to activity changes in response to food scarcity.

Done.

Page 3:

L3: I advise putting keywords in alphabetical order.

Done.

L9: I would recommend mentioning some example mammal populations in brackets.

Done.

L16: I think I understand what you mean by field signs but maybe short elaboration is better. Give an example for instance. I am not sure if field sign is the most appropriate word here.

→ Footprints and feces have been described as examples.

L27: what is the difference between hunting bag and harvest record? I would have expected that it is the same. Thus, I would advise explaining the definition of harvest record already in an early stage since it is also used in the title.

→ We have made the usage of terms being consistent, and clarified the definition.

L41: can you elaborate a bit more which parameters for instance are estimated by the camera trap data so that it is better to understand for a reader without the background knowledge of the REST model?

→ Population density has been added as an example.

L51: You can even put it in a broader context. Not only in Japan but worldwide wild boar are causing damage. You could start with the fact that worldwide wild boar are causing damage with some countries as example and then narrow down to also in Japan they are causing damage and continue your story. That makes it maybe even more interesting for a broader public.

Done.

L52: “As abandonment of agricultural fields increases due to human population aging, sites preferred by wild boar, including abandoned fields and bamboo forests, have been increasing, thereby leading to crop damage near residential areas. This sentence is not really clear to me. It is a long sentence and a bit confusing. I would recommend rephrasing it.

→ This sentence has been split and rephrased to make it clearer (L47-49).

Page 4:

L2: delete “the” in front of “environmental conditions”.

Done.

L2-L9: that paragraph is a bit confusing to me. I think the main points you want to make is that catchability, population size as well as habitat use change spatially and seasonally. Because you mention the reason for two of the factors (habitat use and population size) in between of the enumeration of the three parameters the readers gets a bit confused. I would advise rephrasing this paragraph.

→ As you pointed out, the logical flow of this paragraph was not good and unclear. We have rewritten the paragraph to make it clearer (L47-56).

L17: Why does simultaneous estimation in a Bayesian framework results in less error? I would recommend giving some additional explanation for this. Does not need to be long, but one addition sentence may help to get more clarity here.

→ The following sentence has been added, “because independent estimates can amplify errors by themselves”. (L60-61)

L20: You mention here trapping records. I guess that this is the same as harvest records? But that is a guess and not really clear. I would advise using consistent words for it. And also some additional information about it. I was wondering for instance how sure you are about the fact that trapping by different municipalities and during different seasons

was conducted with a constant effort? If that is an assumption you have to be quite certain about it, because I think that this can have huge impact on the results if it is not conducted with constant effort. Otherwise it might be really similar to hunting bag? I see now that you mention the trapping effort in the methods. That's good, but I would say, make sure that the reader knows here already that you take different trapping effort into account. Simply one short additional sentence should be enough.

→ The following information on the harvest records has been added “mainly number of trapped individuals and trapping effort” in the introduction part (L66).

L24-L29: I would rephrase this last part of the introduction. At the moment it feels a bit lost from the earlier paragraph. I would recommend something like: “Based on seasonal dynamics of the three parameters, we proposed an effective trapping strategy for sites and timing of trap placement on a landscape level for an example study area in the southern part of Chiba prefecture, central Japan.” By simply switching first and last part of your paragraph it better links to the earlier paragraph.

Done

Page 5:

L1: There seems to be a clear distinction between hunting and trapping. I think that this needs to be clearer for the reader at an early stage. I start now understanding that there is a difference, but it can be much easier for the reader if it is explained clearly at the beginning of the manuscript. I think that it namely not so common to have trapping (as you define it) next to hunting methods for wild boar control. At least not in Europe. Thus, I guess that European readers may be a bit confused here.

→ As with the earlier responses, we have made the usage of terms being consistent, and clarified the definition. (L66, 87-88)

L6-L11: I think that you can better combine those two sentences. Now it seems a bit that you rephrase the same sentence again.

Done.

L14: If I understand correctly you distinguished between juveniles (0-4 months) and adults (>4months). Wouldn't you expect differences in catchability and habitat use between piglets (<13 months), sub-adults/yearlings (13-24 months old individuals) and adults (>24 months old individuals) as well? I understand if the distinction of other age classes only on camera traps is not possible but I think that it is important to mention this possible limitation of the study here.

→The limitation of distinguishing yearlings using camera traps has been described, as mentioned in the response to major comment 6.

L21: where the camera traps randomly distributed throughout the area? I guess so because you used the Rowcliffe method, but I don't see that you mention it in this paragraph.

Done

L35: “The environmental variable was based on land use data...” add “was based” to your sentence.

Done

L37: add “average” home range? That is probably more appropriate.

Done.

L45: I think that it is clearer if you write “... six landscape elements...” instead of “six environmental elements”.

Done.

L60: see my comments on the age classes before. I think some additional explanation is necessary why you did not distinguish between yearlings and adults as well. Especially when you have trapping data more than only the two age classes you distinguish may be available.

→ Yearlings are not always distinguishable for some individuals due to rapid body growth in Japan. Even more important is to match the trapping data with camera data.

Page 6:

L9: replace “rest were” by “remaining percentage was”.

Done.

L29: what is the distribution of the staying time? is there a lot of variation? I would like to see a histogram of the staying time variable in the supplementary material.

→ added as a supplementary information (S4).

Page 8:

L17: Put posterior distributions in the supplemental material please.

→ added as a supplementary information (S6).

L28: rather than “environmental variables”, I would name it “landscape variables”. That is more precise and clearer to the reader.

Done.

L39-41: I would put those two lines in the methods instead of in the results.

Done.

Page 9:

L27: “simultaneous estimation”. Is that one of the major advantages of this technique/tool that you used here? If so, I would make that clear to the reader. At the moment it pops up in the title and here in the first sentence of discussion, but to me it is not really clear if that is an important point or not. Needs some clarification here. Ok, I see now in the next sentences of the discussion that you mention the advantages. I would already see those points in the introduction (when elaborating the problem).

→ I have already described a brief explanation in the introduction part.

L29: delete “quite” in sentence “quite useful”. It is useful!

Done.

L40: I miss a bit more in-depth detail about how exactly this methods shows advantages. Why and how are parameter estimates more reliable?

→ We agree “more reliable” is not appropriate here. Now rephrased as follows, “Considering seasonal fluctuations, we seemed to have obtained reasonable estimates of population density”.(L245)

L45: refer to the reports. How do we know what is the “correct” density? How do we know if those estimates are more reliable than others, which are also estimated? That is not explained enough here and should be more convincing.

→ We have removed the phrase of the more precise estimates in comparison to previous reports.

L60: can you add overview of number of cameras in each landscape element in the study area (in supplemental material) so that you can underline your word “mainly” by a number/percentage. That makes your sentence more convincing.

→ “Mainly” was actually incorrect, and reworded as “all”.

Page 10:

L4: ON average. Not “in average”.

Done.

Page 11:

L20: Can you also mention management implications for countries that do not use trapping as control measure for the population?

→ The applicability of our model to any types of hunting data, including shooting data, has described in the paragraph. (L313-316)

L39-L46: As mentioned in my general comments, you can put much more focus on this last paragraph in your introduction and make it much more interesting for a broader public.

→ We have described the general focus of our study, with two sentences, in the introduction part (L23-25, 68-70).

Page 12:

L10: reference nr 17 populationabundance should be two separate words.

Done.

Page 15:

Figure 1: What are the boxes and circles indicating? Mention it in the figure description. I anyway need some addition explanation for figure 1. When you only read the introduction where the first time is referred to figure 1 and by having only those information from introduction and figure description it is difficult to understand the figure. For instance, what is meant with density/mesh? I would advise either give additional explanation in introduction and/or figure description or make an easier graphical representation for the introduction.

→ We have revised the figure and added detailed description in the figure legend to make the meaning clearer. The framework includes estimated parameters (circles) and

data (rectangles) at three different spatial scales (camera site, approximate home range, and management unit). We believe the new figure becomes much easier to understand.

Page 16:

L 50: Figure 4: is juvenile density never below adult density or not visible in month December-April for the juveniles because it lies below the density of the adults? If the latter is the case I would advise to use lines instead of filling the surface. Wouldn't be nicer to add also the bars in the two different colors (light and dark grey) for the two age classes? What is the reason to use the 50% credible interval and not a 95% confidence interval or a standard error?

→ This is actually a stacked area graph, rather than line graph with filled area. No juveniles were observed from December to March. Such explanation has been added in the figure legend. Since we used Bayesian estimation of population density, "credible intervals" are more appropriate. 50% credible intervals are often used in applied studies, such as fisheries management, epidemiology, and climatic predictions. This is because the aims are parameter estimation, not statistical testing. As we want to know seasonal trends in population density and trap catchability, we used 50% credibility intervals. For estimating environmental preference (Fig. 6), we used 95% credible intervals, as we want to know whether forests with a particular land-cover nearby are preferred more than just a chance.

Page 17:

L20: put the x-axis with the month at the bottom (at y=-1 for instance) so that it is easier to read and not all the months names are below the lines and points. This makes the graph a bit confusing.

Done.

L54: "throughout the year" instead of "through a year".

Done.

Supporting information:

L14: typing error: "monitoring".

Done

L20: Fig. S4: b) y axis is entitled with number of videos with two most left graphs, but has frequency on y-axis for the two most right and two central graphs. I think that you should remove the "frequency" on the y-axis, as it is also removed in (a). I don't understand in the figure description what you mean with y-values of circular points in the figures. I don't see any circular points.

→ We have removed the description of circular points.

L26: In reference Rowcliffe et al, (2014) is missing.

Done

Responses to Reviewer: 2

Thank you for valuable comments. **The revised parts are indicated by red letters in the revised manuscript.** Below you can see the responses to each of your comments.

I have found your work timely and interesting. My main concern is the potential pseudoreplication in your experimental design. I think that the incorporation of the camera id as a random term (e.g., intercept), will improve your Bayesian modelling preventing from getting spurious effects (see, Paul-Christian Bürkner. 2020. Package 'brms' 2.12.0 version).

→ Thank you for the comment. We agree that pseudoreplication may be an important issue in the analyses of spatial data. However, the current model should have already considered the possible random effects at camera stations. We assume that the number of videos captured by camera traps follows a negative binomial distribution, which is equivalent to the Poisson distribution with gamma random effects. Although we did not consider the random effects in the staying time, our preliminary analyses based on WAIC model selections suggest that models accounting for the effect were mostly not better than those without it, as shown below. In the manuscript, we added the explanations in Line 152-154, 163.

Table. Values of WAIC (Watanabe-Akaike widely applicable information criterion, Watanabe 2013) after fitting log-normal distribution with and without random effect at camera stations to the staying time. Minimum values are shown in bold.

No.	Age	Year	Month	With random effect	Without random effect
Adult	2018	6	280.0	267.3
Adult	2018	7	292.7	270.4
Adult	2018	8	259.5	264.0
Adult	2018	9	261.1	247.7
Adult	2018	10	251.6	238.5
Adult	2018	11	318.2	293.2
Adult	2019	12	309.5	283.1
Adult	2019	1	289.4	282.7
Adult	2019	2	298.8	289.8
Adult	2019	3	267.9	268.5
Adult	2019	4	275.8	267.8
Adult	2019	5	286.5	276.0
Juvenile	2018	6	259.8	256.5
Juvenile	2018	7	297.8	283.6
Juvenile	2018	8	279.3	263.7
Juvenile	2018	9	272.2	271.2
Juvenile	2018	10	103.2	99.3
Juvenile	2019	5	115.7	112.3

Other minor comments have been included in the pdf.

→ The first comment is not relevant, as this sentence describes wildlife issues in general, and citing references of deer is not out of place. The second comment is about the range expansion of wild boars in urban areas. This has been briefly touched.(L35-36)

Appendix C

Responses to the comments

Thank you for the comments.

I have revised the final manuscript as follows.

Minor comments:

Page 13:

Line 38: make clear that the reader understands the novelty of this approach. You say that it accounts for spatial and temporal heterogeneity in detectability. Maybe it would be good to add one sentence that in the previous approaches, this has not been done. I see that this is what you want to make say, but I would state it in an additional sentence.

→ The following sentence has been added, "Since earlier methods assumed simple situations with no spatial structures of densities,"

Page 17:

Line 20: Put the table that you used as response to the comment from reviewer 2 in the supplement as well and refer to it here.

→ The table has been added to the supplementary file (table S1).

Page 22:

Line 27: What do you mean with "in separation"? seperately is maybe a better word?

→ Changed as "with single density proxy or multiple proxies used separately"

Line 29: You say that the density estimates become much more precise. Is it precise? Or do you mean accurate? You state it but you did not test (statistically) if the results are more precise/accurate right? I would rephrase this sentence a bit to reduce the chance of misinterpretation by the reader.

→ Changed to "unbiased"

Line 30: Off course our models can also be used for eradication...". Add also in the sentence for clarity.

→ "Off course" has been deleted and "Other than species conservation" has been added instead.